# Striatum-projecting prefrontal cortex neurons support working memory maintenance

Maria Wilhelm[1,2,6,10], Yaroslav Sych [1,7,10], Aleksejs Fomins[1,2], José Luis Alatorre Warren[1,8], Christopher Lewis [1], Laia Serratosa Capdevila [1], Roman Boehringer[3], Elizabeth A. Amadei [3], Benjamin Grewe[2,3,4], Eoin C. O'Connor [5], Benjamin J. Hall[5,9] & Fritjof Helmchen [1,2,4] ✉

Neurons in the medial prefrontal cortex (mPFC) are functionally linked to working memory (WM) but how distinct projection pathways contribute to WM remains unclear. Based on optical recordings, optogenetic perturbations, and pharmacological interventions in male mice, we report here that dorsomedial striatum (dmStr)-projecting mPFC neurons are essential for WM maintenance, but not encoding or retrieval, in a T-maze spatial memory task. Fiber photometry of GCaMP6m-labeled mPFC→dmStr neurons revealed strongest activity during the maintenance period, and optogenetic inhibition of these neurons impaired performance only when applied during this period. Conversely, enhancing mPFC→dmStr pathway activity—via pharmacological suppression of HCN1 or by optogenetic activation during the maintenance period—alleviated WM impairment induced by NMDA receptor blockade. Moreover, cellular-resolution miniscope imaging revealed that >50% of mPFC→dmStr neurons are active during WM maintenance and that this subpopulation is distinct from neurons active during encoding and retrieval. In all task periods, neuronal sequences were evident. Striatum-projecting mPFC neurons thus critically contribute to spatial WM maintenance.

Working memory (WM) is the ability to temporarily hold and manipulate relevant information in one's memory to guide future actions, a process that is essential for many forms of goal-directed behavior. The medial prefrontal cortex (mPFC), which is densely connected to many other brain regions, has a central function in executing tasks that require WM[1–10]. Indeed, dysfunctions of mPFC, as they occur in several mental disorders including schizophrenia, are associated with WM deficits[11–13]. Despite extensive research on mPFC, we still incompletely understand how mPFC neurons contribute to WM and what type of

information they transfer via specific pathways to downstream target areas[14].

Neurons in the mPFC receive inputs from a diverse set of brain structures and their major axonal projections include the mediodorsal nucleus (MD) in the thalamus, the dorsomedial striatum (dmStr), the basolateral amygdala (BLA), and the ventral tegmental area (VTA)[14]. Recent studies have begun to dissect the specific WM contributions of afferent and efferent pathways in this complex network. For example, projections from hippocampus to mPFC in mice were found to be

[1]Brain Research Institute, University of Zurich, 8057 Zurich, Switzerland. [2]Neuroscience Center Zurich, University of Zurich and ETH Zurich, 8057 Zurich, Switzerland. [3]Institute of Neuroinformatics, University of Zurich and ETH Zurich, 8057 Zurich, Switzerland. [4]University Research Priority Program (URPP) Adaptive Brain Circuits in Development and Learning (AdaBD), University of Zurich, Zurich, Switzerland. [5]Neuroscience & Rare Diseases, Roche Pharma Research and Early Development, Roche Innovation Center Basel, F. Hoffmann-La Roche Ltd, Basel, Switzerland. [6]Present address: Institute for Neuroscience, ETH Zurich, 8057 Zurich, Switzerland. [7]Present address: Institute of Cellular and Integrative Neuroscience, CNRS, University of Strasbourg, Strasbourg, France. [8]Present address: Center for Lifespan Changes in Brain and Cognition, University of Oslo, Oslo 0317, Norway. [9]Present address: Circuit Biology Department, H. Lundbeck A/S, Valby, Denmark. [10]These authors contributed equally: Maria Wilhelm, Yaroslav Sych. ✉e-mail: helmchen@hifo.uzh.ch

critical for encoding, but not maintenance or retrieval, of spatial cues in a WM task[15]. In addition, analysis of mPFC interactions with MD in the thalamus revealed that activity of the MD→mPFC pathway is essential for sustaining prefrontal activity during WM maintenance[15–17]. The reciprocal mPFC→MD pathway, on the other hand, was found to mainly guide successful WM retrieval and to support subsequent choice[16]. This efferent pathway appears less important for WM maintenance during the delay period, when MD activity actually leads activity in the mPFC[16], and it is not essential for encoding. A further dissection of the functional roles of specific pathways emerging from mPFC (or targeting mPFC) is needed to puzzle together the precise involvement of mPFC in WM, and especially WM maintenance.

Here, we hypothesize that the projection from mPFC to dmStr is a likely candidate pathway to support maintenance of information in a WM task. Consistent with this hypothesis, both the prelimbic (PrL) and infralimbic (IL) regions of mPFC contain neurons that project to the anterior region of dmStr[18,19]. Lesions of dorsomedial, but not dorsolateral, striatum result in WM impairments[20,21]. In addition, electrophysiological recordings from either mPFC or dmStr revealed neuronal populations that display activity patterns spanning WM maintenance periods in a sequential manner[16,22–24]. This pathway may have multiple overlapping roles related to WM maintenance, overall control of premature actions, and impulsive behaviors, including the necessary control of behavioral inhibition[25] in tasks that involve delayed action. Therefore, we consider it likely that the mPFC→dmStr pathway is part of the brain circuitry that is essential for delayed goal-directed behavior, albeit it remains unclear what exact information is conveyed between these two regions. To specifically probe the function of this pathway in a spatial memory task, we performed fiber-optic calcium recordings of mPFC→dmStr neuronal activity as well as cellular-resolution miniscope imaging to analyze activity patterns in striatum-projecting mPFC neurons. Using optogenetic and

pharmacological manipulations, we tested the functional significance of mPFC→dmStr pathway activity during specific periods of the WM task. Our results corroborate the notion that the mPFC→dmStr pathway is critically involved in WM maintenance periods.

## Results

### Fiber photometry of mPFC→dmStr pathway activity in a spatial memory task

We trained C57BL/6 mice in a T-maze delayed non-match-to-place (DNMTP) task commonly used to study spatial WM[26–28]. In this task, freely moving mice receive a water reward during a 'choice run' when they correctly choose the left or right arm of the T-maze that is opposite to the arm they visited in the previous 'sample run' ("Methods"). Each trial consists of three periods related to the *encoding (E)*, *maintenance (M)*, and *retrieval (R)* of WM (Fig. 1a). In the encoding period (sample run), one of the two T-maze arms is blocked by a door and the mouse is directed towards a first water reward in the open arm. After turning back towards the start box at the end of the sample run and throughout the subsequent delay period, the mouse must maintain information about the location of the sample reward in WM. In our setup, we enforced a delay waiting period in the start box of at least 5 seconds (5 s for fiber photometry and miniscope experiments; 10 s for optogenetic and pharmacological perturbations; "Methods"). Finally, in the choice run, all doors open and the mouse must retrieve the information held in WM in order to correctly alternate to the T-maze arm previously not visited. Correct choices result in a second water reward, triggered by licking at the waterspout. Upon a mistake, no second reward is delivered, and the mouse has to return to the start box to initialize the next trial. After completing training ("Methods"), mice typically reached performance levels of 70–80% correct trials. The direction of turn during the sample run was chosen pseudo-randomly for every trial, therefore mice could not learn this task by a

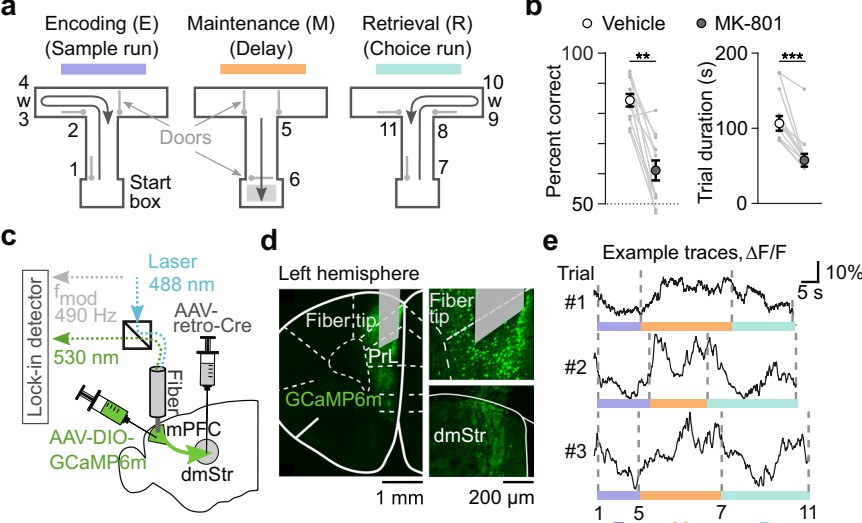

**Fig. 1 | Fiber photometry of calcium signals in striatum-projecting mPFC neurons during T-maze behavior. a** T-maze DNMTP task with automated doors to enforce sample runs and enable choice runs. Numbers indicate salient task events: 1–start of the sample run, 2–turning at the T-junction of the main maze arm, 3–first water reward, 4–end of licking period, 5–turn to run towards the start box, 6–reaching the start box, 7–start of the choice run, 8–turning at the T-junction, 9–second water reward, 10–end of licking period, 11–end of choice run. The second water reward is omitted in incorrect trials. Encoding (1–5), maintenance (5–7), and retrieval (7–11) periods are colored in purple, orange, and cyan, respectively. **b** Left: Task performance is impaired by the NMDAR-blocker MK-801 (0.1 mg/kg, i.p.), **p = 0.002, two-sided Wilcoxon signed rank test. Right: MK-801 reduced mean trial duration significantly, ***p = 9.76 × 10⁻⁴, two-sided Wilcoxon signed rank test. Data

are shown as mean ± s.e.m., individual mice (n = 11) are represented with gray lines. **c** Schematic of virus injections and fiber implantation in the left hemisphere for photometric calcium recordings of dmStr-projecting mPFC neurons. Laser excitation at 488 nm excitation light and detection of GCaMP6m green fluorescence are depicted schematically. **d** Left: Cre-dependent expression of GCaMP6m in a coronal histological section with implanted fiber tip shown above the prelimbic area (PrL). Right, Confocal images of GCaMP6m-labeled cell bodies in PrL (top, +1.8 AP, "Methods") and axonal terminals in dmStr (bottom, +1.05 AP, "Methods"). **e** Example ΔF/F traces of pathway-specific photometric GCaMP6m recordings in PrL for 3 correct trials. Dashed vertical lines and purple, orange and cyan horizontal bars indicate the different task periods according to panel a. Note the variable duration of trial periods across trials. Source data are provided as a Source Data file.

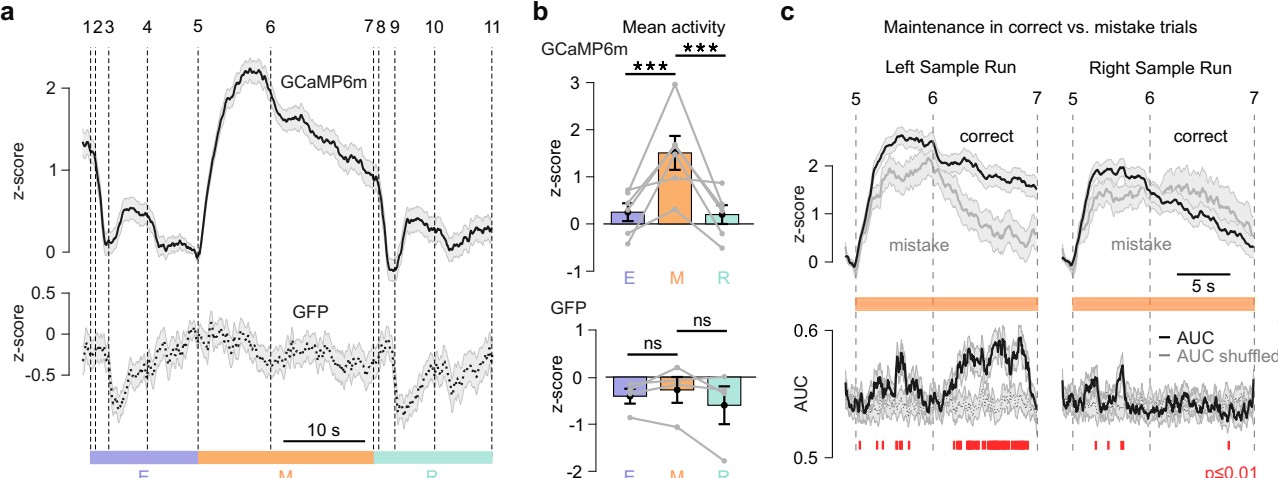

**Fig. 2 | mPFC→dmStr activity correlates with WM maintenance. a** Resampled and session-averaged z-scored calcium signal ($n = 6$ mice; solid line) and non-calcium dependent GFP control signal ($n = 4$ mice; dashed line). Vertical dashed lines and numbers indicate trial events. Shaded areas indicate s.e.m. **b** Mean z-score values ± s.e.m. over the different task epochs. Each gray line represents one mouse (***$p = 7.71 \times 10^{-15}$ for E vs M, ***$p = 1.15 \times 10^{-15}$ for M vs R, $p = 0.99$ for E vs R; $p$-values combined for mice using Fisher's method; $n = 6$ mice; 103–156 correct trials per mouse; for individual mice, we performed one-way ANOVA and Tukey post hoc test for multiple-group comparisons). ns, non-significant. **c** Top: Mean z-scored calcium signals during the maintenance period for correct alternation in choice runs (black) and for mistakes (gray). Data are shown separately for periods following left sample runs (correct choice to the right, contra-lateral to the recording site) and right sample runs (correct choice to the ipsi-lateral side). Bottom: Time course of correct vs. mistake classification based on the area-under-the-curve (AUC) of an ROC analysis. Red markers at the bottom indicate time bins with significant (adjusted $p$-value ≤ 0.01) classification accuracy compared to trial-shuffled data (gray trace). Individual $p$-values corresponding to each red line (each time bin) are indicated in the Source Data file and were calculated by applying two-sided Wilcoxon rank sum test to each time-bin comparison followed by false discovery rate correction (Benjamini-Yekutieli). Solid lines represent mean, shaded area ± s.e.m. Source data are provided as a Source Data file.

simple alternation rule. In addition, we performed a detailed analysis of mouse behavior during the maintenance period based on video tracking (see "Methods"). We found no evidence that any behavioral variable (e.g., mouse movement, direction and frequency of turns, head direction etc.) was predictive of the future left and right turn at the T-junction decision point (see Supplementary Fig. 1 for variability across mice and behavioral analysis).

Pharmacological administration of the NMDA-type glutamate receptor (NMDAR)-blocker MK-801 is known to impair specifically WM[29,30]. We confirmed these results[26,31] and verified that acute systemic application of MK-801 (0.1 mg/kg) significantly reduced T-maze DNMTP task performance and shortened trial durations across all periods (Fig. 1b), indicating hyperlocomotion. Importantly, application of MK-801 did not affect the overall behavior in the maze, that is mice still correctly completed the general task sequence, initiating licking on the waterspout and returning to the start box. The effect of MK-801 cannot be attributed to a specific neural pathway, though.

To specifically measure the activity of mPFC→dmStr projection neurons during behavior, we injected retrograde Cre-expressing virus unilaterally into the left dmStr and a Cre-dependent GCaMP6m-expressing virus into left mPFC, targeting the prelimbic area PrL ($n = 6$ mice; "Methods"). To perform fiber photometry, we chronically implanted an optical fiber above the mPFC injection site (Fig. 1c, d). Once mice had reached expert level (≥60% correct trials on 2 consecutive days), we performed pathway-specific calcium recordings while mice executed the DNMTP task. Relative percentage changes in fluorescence (ΔF/F) showed trial-related activity with transient changes up to 30% amplitude and variable time courses across trials and mice, indicating spiking activity of mPFC→dmStr projection neurons (Fig. 1e).

### The mPFC→dmStr pathway sustains high activity during WM maintenance

Based on 11 salient trial-related events we defined 10 trial phases (Fig. 1a). Because of freely-moving behavior, the duration of these

phases varied across trials, including the WM maintenance period. The maintenance period was defined according to the idea, that the WM is required to temporarily store information which is not available to the immediate sensory input. Therefore, the maintenance period started at event 5 (Fig. 1a; turning back towards the start box, the full body of the mouse is in the corridor towards the Start-box) and ended at event 7 (start of the choice run, the mouse is moving towards the T-junction) (Fig. 1e). To analyze the temporal profile of mPFC→dmStr activity across trials, sessions, and animals we had to account for the variability of trial phase duration. To this end, we aligned the photometric calcium signals to the 10 salient trial phases by segment-wise resampling the ΔF/F traces so that each phase duration matched the median of the corresponding phase duration across all trials ("Methods"). We then averaged the resampled and z-scored fluorescence signals for 6 GCaMPm-expressing mice and 4 GFP-expressing control mice ($n = 5$ sessions for each mouse). The mPFC→dmStr pathway exhibited the highest activity during the WM period, with a strong signal increase at the beginning of the return to the start box. The signal remained high, albeit slowly declining, until the start of the choice run (Fig. 2a). The mean activity in the maintenance period was significantly higher compared to both retrieval and encoding period (Fig. 2b; $p < 0.001$, Fisher's method).

We tested whether calcium signals might be confounded by motion artefacts by recording control ΔF/F signals excited at 425-nm wavelength, at which GCaMP6m has low calcium-sensitivity (simultaneously with 488-nm excitation; "Methods"). Fluorescence signals excited at 425 nm displayed substantially lower variation compared to 488-nm excited signals (Supplementary Fig. 2a, b). In addition, z-scored fluorescence traces in the control GFP mice were relatively flat with no difference in mean activity between trial periods (Fig. 2b). Small fluorescence decreases that were apparent in phases 3–4 and 9–10 possibly originated from animal movements or hemodynamic signals associated with reward consumption, which does not, however, confound our analysis of the maintenance period. We also analyzed multiple behavioral variables but did not find clear clusters of distinct behaviors, rather a spectrum of behaviors (Supplementary Fig. 2c). To

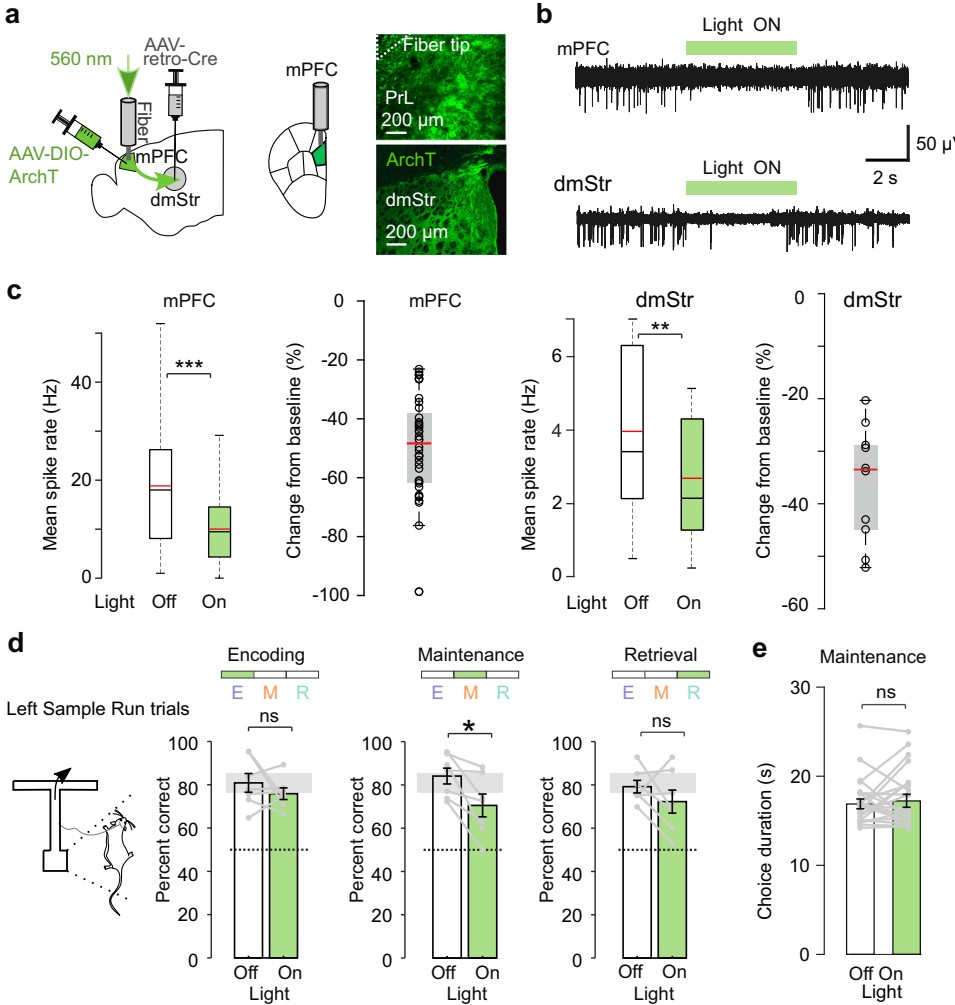

**Fig. 3 | Photoinhibition of mPFC→dmStr activity during WM maintenance impairs task performance. a** Left: Schematic diagram of viral injections and optical fiber implantation in the left hemisphere for pathway-specific ArchT expression and photoinhibition of mPFC→dmStr projection neurons. Right: confocal images of ArchT-expressing cells in PrL below the fiber tip (top) and ArchT-expressing axons in dmStr (bottom). **b** Example of multi-unit recordings from cells in mPFC (top) and dmStr (bottom) showing suppressed activity during 560-nm illumination of ArchT-expressing mPFC neurons (indicated by green bars). **c** Light-induced suppression of multi-unit activity in mPFC (left) and dmStr (right). Box-whisker plot of mean spike rates for Light Off/On conditions shows minimum-maximum range and 25th and 75th percentiles (black line: median; red line: mean; ***$p = 3.6 \times 10^{-8}$, 40 sites from $n = 3$ mice in mPFC; **$p = 0.002$, 10 single sites from $n = 2$ mice in dmStr; two-sided Wilcoxon signed rank test). In addition, percentage changes of spike rate relative to Light Off condition are shown (red line: population

mean; gray box: 25th/75th percentiles; whiskers: minimum-maximum range; circles: single sites on the electrophysiological probe). **d** Optogenetic inhibition of the mPFC→dmStr pathway during the encoding, maintenance, and retrieval periods. Performance was reduced during the maintenance period from 84.3 ± 3.7% to 70.7 ± 5.3%; Left Sample Run trials; mean ± s.e.m; $p(E) = 1.0$, *$p(M) = 0.047$, $p(R) = 1.0$ (two-sided Wilcoxon signed rank test with Bonferroni correction); Right Sample Run trials performance did not show any significant effects; $p(E) = 1.0$, $p(M) = 1.0$, $p(R) = 1.0$ (two-sided Wilcoxon signed rank test with Bonferroni correction). Gray lines represent individual mice ($n = 7$ mice). Gray shaded error bars represent the s.e.m. spread during the Light Off encoding condition. **e** Lack of effect of photo-inhibition on the choice duration. Each line represents average choice duration (both contra- and ipsilateral turn trials) during individual sessions ($p = 0.88$, $n = 7$ mice, 3 sessions per mouse; two-sided Wilcoxon signed rank test). Source data are provided as a Source Data file. ns, non-significant.

evaluate whether variability of mPFC→dmStr activity during the maintenance period could be explained by this behavioral repertoire, we analyzed mPFC→dmStr activity at the extremes of the behavioral spectrum but did not find any obvious relationship between activity and the ongoing motor behavior (Supplementary Fig. 2d–g).

We further assessed whether mPFC→dmStr activity differs for correct versus mistake trials. Because the activity of mPFC neurons may predict the future reward location (left versus right goal arm)[24,32], we further stratified the fluorescence signals—recorded in the left hemi-sphere—into contra-lateral (right) and ipsi-lateral (left) choices. Indeed, fluorescence signals during the maintenance period were larger for correct choices towards the contralateral side compared to correct

ipsilateral choices (Supplementary Fig. 3). For contralateral choice runs, mPFC→dmStr activity was significantly enhanced in the maintenance period when the animal's turn was correct compared to mistake trials (Fig. 2c). These results are in line with previous findings that the activation of striatal neurons in one hemisphere precedes the initiation of contralateral movements[33]. The most significant difference between correct and mistake trials occurred in the late maintenance phase for contralateral choices (Fig. 2c; second half of the phase between events 6 and 7; quantified by a receiver operating characteristics (ROC) analysis; "Methods"). These findings suggest that the performance in the DNMTP task may critically depend on the mPFC→dmStr pathway activity specifically during the WM maintenance period.

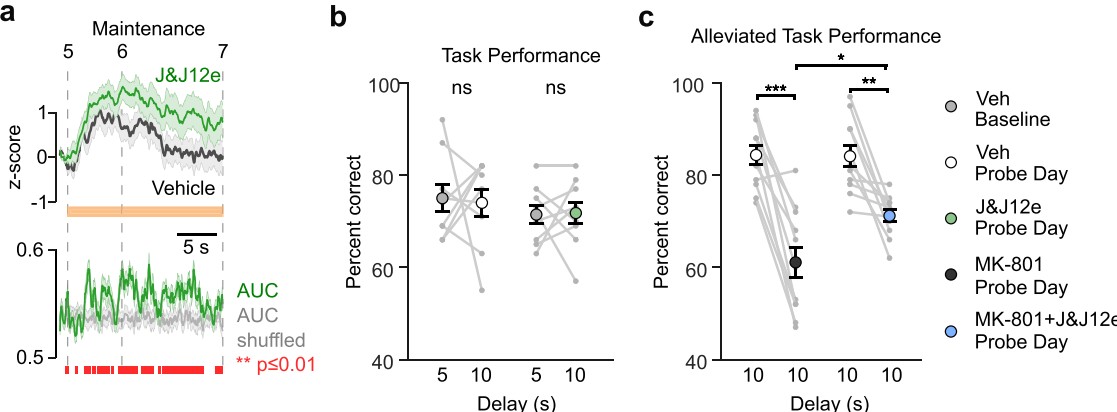

**Fig. 4 | Blockade of HCN channels enhances WM-related mPFC→dmStr activity and rescues WM impairment induced by MK-801. a** Pathway-specific activity is higher when mice receive the HCN-channel blocker J&J12e (green, 3 mg/kg, i.p.) compared to vehicle injection (gray). Average of z-scored ΔF/F traces for all correct trials (*n* = 6 mice). Bottom: Time course of classification power showing AUC of an ROC analysis. Red markers below indicate time bins with significant (adjusted *p*-value ≤ 0.01) classification accuracy compared to trial-shuffled data (gray trace). Individual *p*-values corresponding to each red line (each time bin) are indicated in the Source Data file and were calculated by applying two-sided Wilcoxon rank sum test to each time-bin comparison followed by false discovery rate correction (Benjamini-Yekutieli). Solid lines represent mean, shaded area ± s.e.m. **b** No effect on performance was observed for both 5-s and 10-s delay conditions (one-way ANOVA and Tukey post hoc test for multiple-group comparisons; all exact p-values are indicated in the Source Data file). ns, non-significant. **c** Impairment of task performance by NMDAR blockade with MK-801 (left, 0.1 mg/kg, i.p.) was alleviated by additional administration of J&J12e (right) as indicated by a smaller drop in percentage of correct trials. ***p(Veh vs MK801) = 3.54 × 10⁻⁷; *p*(Veh baseline day1 vs Veh baseline day2) = 0.99, *p(MK801 vs MK801 + J&J12e) = 0.0310, **p(Veh vs MK801 + J&J12e) = 0.0036; one-way ANOVA and Tukey post hoc test for multiple-group comparisons. Data are shown as mean ± s.e.m., individual mice (*n* = 11) are represented with gray lines. Source data are provided as a Source Data file.

### mPFC→dmStr pathway activity is required for WM maintenance

Guided by our photometry results, we next aimed to test whether pathway-specific optogenetic perturbation of neural activity during the maintenance period would lead to changes in the task performance. First, we tested if mPFC→dmStr activity is required for WM maintenance using optogenetic silencing. To selectively inhibit this pathway, we injected retrograde AAV-Cre into the left dmStr and Cre-dependent AAV driving expression of the light-driven proton pump archaerhodopsin ArchT[34] into the left PrL. This approach resulted in strong ArchT expression in striatum-projecting PFC neurons (Fig. 3a; *n* = 7 mice; "Methods"). ArchT has been previously applied to silence mPFC neurons[15,16,35,36] but for further validation we verified in 7 ArchT-expressing mice that 560-nm illumination indeed reduced multi-unit activity in both mPFC and downstream dmStr (Fig. 3b, c; Supplementary Fig. 4). In expert mice, we then transiently suppressed mPFC→dmStr activity during task performance by temporally restricting laser illumination to one of the three WM periods (encoding, maintenance, or retrieval). In each session, green light was delivered in 50% of randomly interleaved trials and targeted to only one of the WM periods. Optogenetic silencing during the maintenance period resulted in a significant performance decrease of 9%, whereas no significant behavioral effect was induced by silencing during the encoding or retrieval period (Fig. 3d; silencing in the left hemisphere during the maintenance period affected task performance only for the Left Sample Runs, when the animal turned right at the T-junction during the retrieval period). Illumination during the maintenance period did not affect the mean duration of choice runs (Fig. 3e), indicating that the performance decrease was not simply induced by altered locomotion. These findings are consistent with the results of our photometry recordings and corroborate the notion that mPFC→dmStr pathway activity is required for WM maintenance.

### mPFC→dmStr pathway activation alleviates WM impairment induced by MK-801

Given that elevated mPFC→dmStr activity positively correlates with task performance and that silencing of this pathway impairs WM, we next sought to establish whether enhancing mPFC→dmStr activity can improve performance or alleviate WM deficits. In a first approach, we assessed whether striatum-projecting neurons are affected by pharmacological blockade of HCN1 channels, an intervention that has been found to improve WM by enhancing mPFC neuronal activity during WM delay periods[37–39]. Here, we applied the HCN-channel blocker J&J12e that readily passes the blood-brain barrier[40]. After training mice in the DNMTP task with a 5-s delay, we tested performance with a challenging 10-s delay after systemic injection of either J&J12e or vehicle. Photometry revealed a significant enhancement of mPFC→dmStr activity (normalized to the 1-s pre-maintenance time window) after mice received J&J12e (Fig. 4a; the effect was not fully specific to the maintenance period, though; Supplementary Fig. 5). This finding indicates that HCN blockade may increase WM-related activity of mPFC→dmStr projection neurons. Despite the elevated mPFC→dmStr activity that we observed during the maintenance period in J&J12e-treated mice, these mice did not show improved performance (Fig. 4b). Hence, to test whether high baseline performance may have masked a potential compound effect, we also applied J&J12e after impairing WM by systemic injection of MK-801. Indeed, co-administration of J&J12e alleviated the WM impairment induced by MK-801 (Fig. 4c; Supplementary Fig. 5).

Systemic injection of an HCN-blocker presumably has broad and nonspecific effects on various brain regions. To test more specifically whether enhancing mPFC→dmStr pathway activity can improve task performance, we additionally took a second approach by inducing pathway-specific expression of channelrhodopsin-2 (ChR2) (Fig. 5a). We validated that transient continuous blue light illumination for ChR2 activation indeed induced action potentials in PrL neurons (Fig. 5b and Supplementary Fig. 6). ChR2 activation during the encoding and retrieval periods had no significant behavioral effect, whereas illumination during the maintenance period resulted in only a small performance increase of about 3% (Fig. 5c; illumination in GFP control mice had no significant effect; Supplementary Fig. 7). Again, we reasoned that the high (~70%) performance of untreated expert mice may have masked any further improvement. We therefore repeated ChR2-mediated mPFC→dmStr pathway activation but this time after pharmacological application of MK-801, which caused a WM deficit that manifested itself as lower baseline performance (below 70%). Under

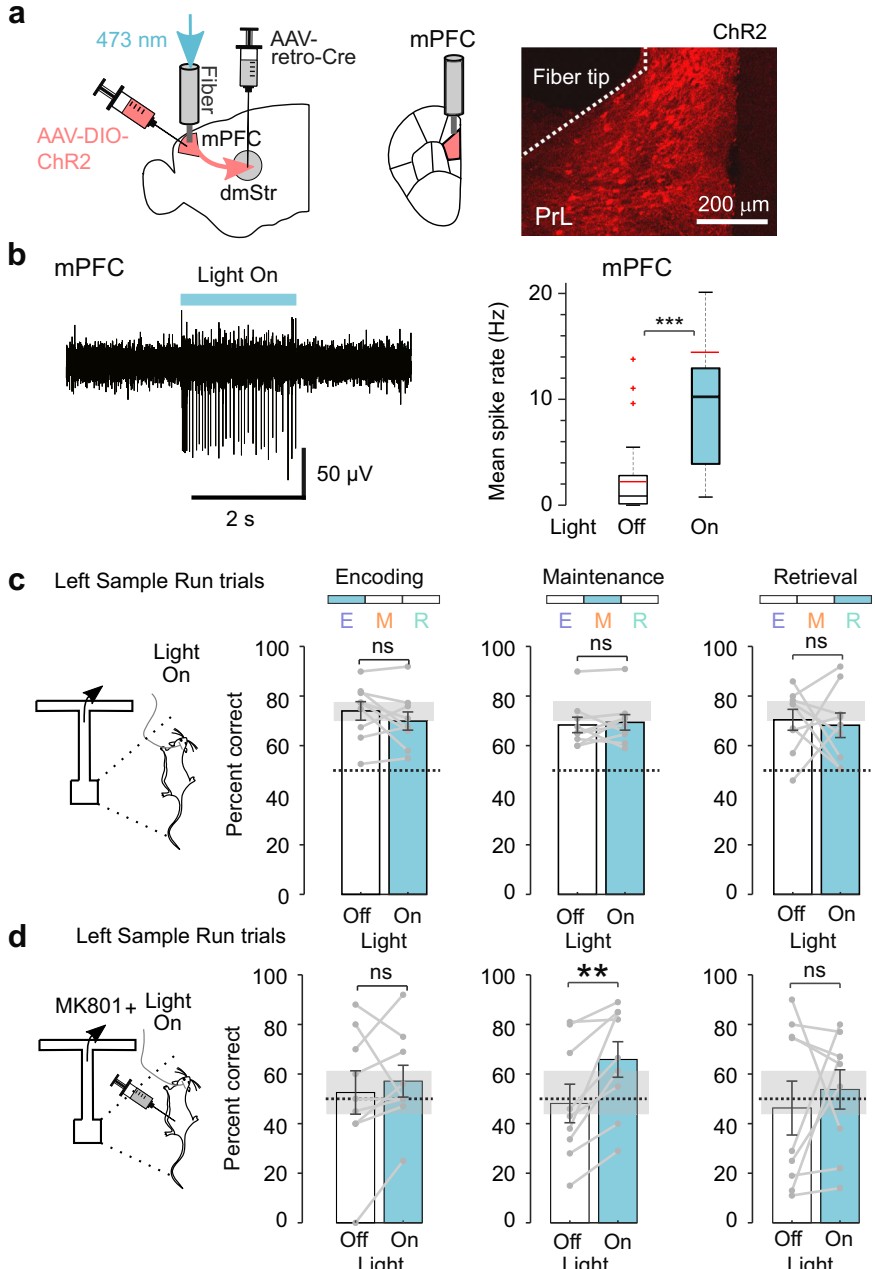

**Fig. 5 | WM effects of optogenetic activation of the mPFC→dmStr pathway.**
**a** Virus injections to induce ChR2 expression in the mPFC→dmStr pathway and fiber implantation site in the left hemisphere. **b** Validation of pathway-specific activation induced by ChR2 activation. Left: example electrophysiological trace from a single site on the electrophysiological probe implanted in PrL; Right: Box-whisker plot of mean spike rates for Light Off/On conditions shows minimum-maximum range and 25th and 75th percentiles (black line: median, red line: mean, red crosses: outliers; ***$p = 7.95 \times 10^{-7}$, 32 units from $n = 3$ mice in mPFC; two-sided Wilcoxon signed rank test). **c** Optogenetic activation of the mPFC→dmStr pathway during the encoding (E), maintenance (M), and retrieval (R) periods in Cre-dependent ChR2-expressing mice ($n = 9$ mice) did not change performance during any period of the task (Bonferroni corrected $p$-values for Left Sample Run trials: E: 0.9 M: 1.0, R: 1.0; Right Sample Run trials performance also did not change in all periods of the trial:

$p$(E) = 1.0, $p$(M) = 0.16, $p$(R) = 0.38 (two-sided Wilcoxon signed rank test with Bonferroni correction). **d** Optogenetic activation of the mPFC→dmStr pathway partially rescued WM impairment induced by MK-801 injection when applied during the WM maintenance period (performance increased from 48.1 ± 7.8% to 65.9 ± 7.2%; Left Sample Run trials, $p$(E) = 1.0, *$p$(M) = 0.011, $p$(R) = 1.0 (two-sided Wilcoxon signed rank test with Bonferroni correction); Right Sample Run trials performance also did not change in all trial periods; $p$(E) = 1.0, $p$(M) = 1.0, $p$(R) = 1.0 (two-sided Wilcoxon signed rank test with Bonferroni correction). For a given task period, ChR2 activation was performed in 50% of trials in a random fashion. Black lines represent individual sessions. Error bars show s.e.m. Gray shaded error areas represent the s.e.m. spread during Light Off encoding condition. Source data are provided as a Source Data file. ns, non-significant.

these conditions, ChR2 activation increased performance, but only if applied during the WM maintenance period (Fig. 5d). Optogenetic activation of the mPFC→dmStr pathway thus partially alleviated MK-801-induced WM impairment, similar to the systemic administration of an HCN blocker.

## Sequential activity in subpopulations of mPFC→dmStr projection neurons

Given that the mPFC→dmStr pathway is most active during the WM maintenance period and because optogenetic manipulations during this period affect behavior, we next asked whether a specific

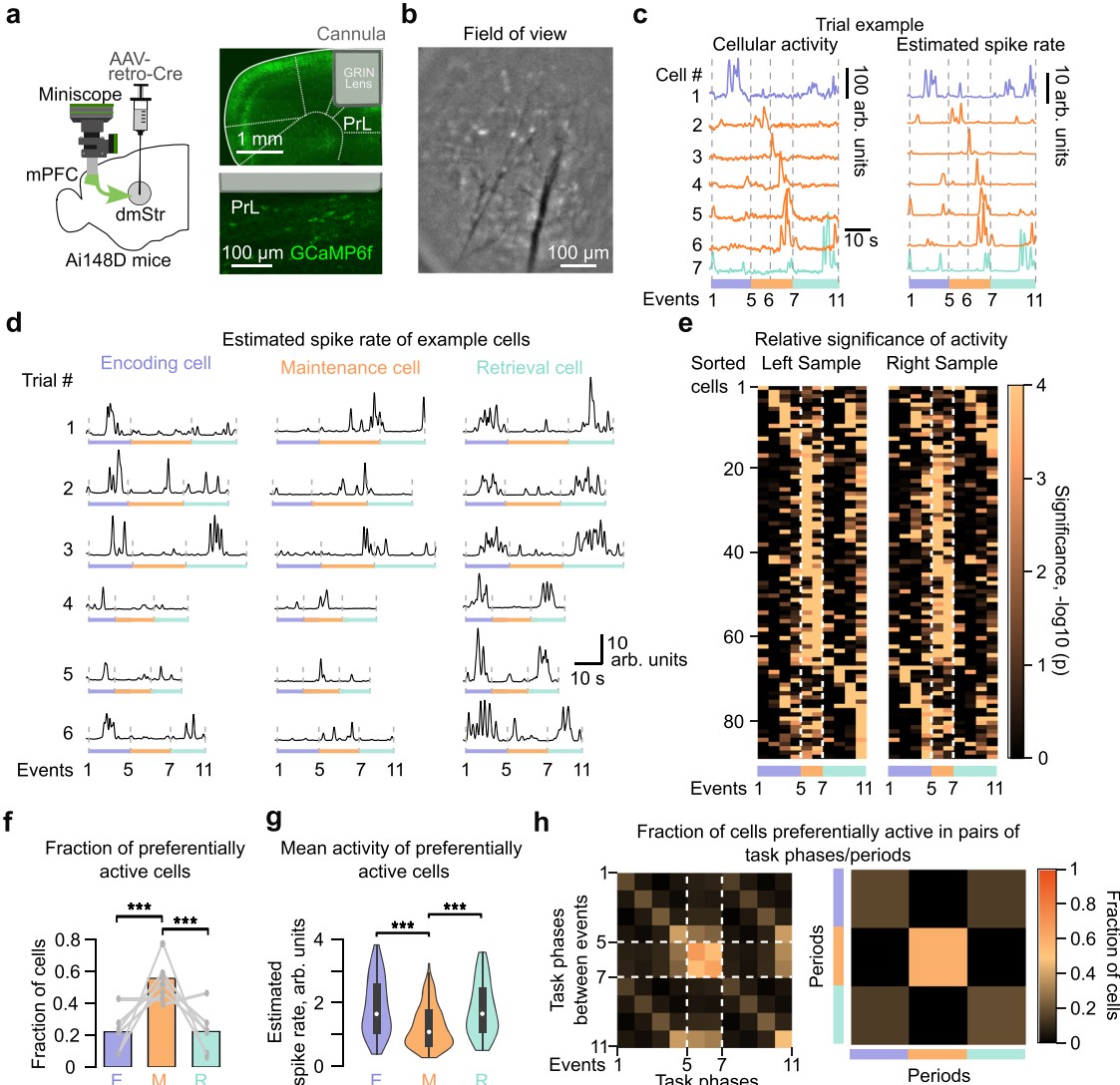

**Fig. 6 | Miniscope calcium imaging of mPFC→dmStr projection neurons during T-maze DNMTP task. a** Left: Schematic of the virus injection and miniscope mounting for cellular resolution imaging of mPFC→dmStr projection neurons. Right: Confocal images of GCaMP6f expression in left PrL indicating GRIN lens insertion into an implanted cannula. **b** Representative field of view for miniscope imaging (single time frame of pre-processed data). **c** Trial-related cellular activity traces of 7 example neurons for one correct trial. Left: output signal in arbitrary units, right: corresponding deconvolved estimated spike rate. Dashed lines, purple, orange and cyan bars indicate different task periods. **d** Example cells identified as being significantly more active in either encoding (left), maintenance (middle), or retrieval (right) period. Six example correct trials are plotted for each neuron. **e** Relative significance of single-cell activity during the 10 trial phases. High values indicate significantly elevated activity of a given cell in a particular trial phase, compared to the rest of the phases. The heatmap shows all cells for one example mouse. Only correct trials were analyzed and cells sorted according to the Left Sample trials. **f** Fraction of cells preferentially active in encoding (E), maintenance

(M), and retrieval (R) period (E: $0.22 \pm 0.05$ M: $0.56 \pm 0.05$, R: $0.22 \pm 0.06$, mean ± s.e.m.; ***$p = 3.97 \times 10^{-12}$ for E vs M, ***$p = 4.01 \times 10^{-12}$ for E vs M, binomial test, p-values composed over mice using Fisher's method with Bonferroni correction; $n = 6$ mice). **g** Mean activity levels of neurons preferentially active in encoding (E), maintenance (M), and retrieval (R) period (E: $0.18 \pm 0.011$ M: $0.123 \pm 0.004$, R: $0.176 \pm 0.01$, mean ± s.e.m.). Mean spike rate for a given period was calculated in arb. units for each trial and averaged for each neuron across all correct trials (***$p = 6.29 \times 10^{-6}$ for E vs M, ***$p = 2.83 \times 10^{-5}$ for M vs R, $p = 0.81$ for E vs R, two-sided Wilcoxon signed rank test, Bonferroni corrected, $n = 6$ mice). Box-whiskers plot shows minimum-maximum range and 25th and 75th percentiles. White circles represent means. **h** Matrix plot showing the fraction of neurons significantly active in pairs of task phases (left) or periods (right) ($n = 6$ mice). Values on the diagonal represent fraction of neurons significantly active in each respective phase or period. Note the overlap of neuronal fractions that show significant activity in encoding and retrieval periods as well as in corresponding phases during these periods. Source data are provided as a Source Data file.

subpopulation of mPFC neurons might exhibit WM-related activity. To achieve cellular resolution we used a wearable miniaturized microscope (miniscope)[41,42] and imaged mPFC neurons through a GRIN lens inserted into a chronically implanted cannula (Fig. 6a; "Methods"). We specifically labeled mPFC→dmStr projection neurons with GCaMP6f, employing the same dual-virus strategy as for fiber photometry in two mice. In addition, we achieved pathway-specific GCaMP6f expression by injecting retrograde Cre-expressing virus into dmStr of four transgenic Ai148D mice (Fig. 6a). Because these two labeling approaches

yielded similar results, we pooled data across all six mice for analysis. Typically, we identified 30–90 longitudinally active neurons within a field of view (Fig. 6b; "Methods").

We observed large task-related calcium transients, with example neurons showing preferential activation in distinct task periods (Fig. 6c). To test for consistency with the photometry experiments, we averaged task-related activity across all active mPFC→dmStr neurons within each FOV. These 'bulk' activity signals resembled the photometric signals, again revealing highest signal amplitude during the

maintenance period as well as significantly elevated activity during the late delay period in correct versus mistake contralateral choice runs (Supplementary Fig. 8). For a more accurate representation of the time course of single-cell activities, we deconvolved neuronal ΔF/F traces using a novel spike inference algorithm[43] (Fig. 6c). All subsequent analysis used the deconvolved traces (expressed in arbitrary units, 'arb. units', because calibration in terms of absolute spike rates is uncertain). First, we analyzed how individual neuron activity relates to the task phases as defined in Fig. 1. Whereas some example neurons showed preferential activation during specific phases of the encoding or retrieval period, others were most active during the maintenance period (Fig. 6d).

To quantify the notion of preferential activity in distinct task periods, we calculated for each active neuron the significance level of it being active in each of the 10 trial phases (90–238 trials per mouse; "Methods"). Sorting neurons according to their most significant phase revealed that distinct subpopulations of neurons were preferentially active during the maintenance period and during the encoding/retrieval periods, respectively (Fig. 6e for an example mouse; Supplementary Fig. 9 for all mice). The highest fraction of active mPFC→dmStr neurons occurred in the maintenance period (Fig. 6f) but the average activity per neuron was slightly lower during maintenance compared to encoding and retrieval (Fig. 6g). Taken together, this leads to an estimate of an 1.74-fold increase in 'bulk' activity during maintenance, which is consistent with the mean activity profile obtained by averaging across all active neurons. The large 'bulk' signals during maintenance thus predominantly reflect the relatively strong recruitment of mPFC→dmStr neurons during this task period.

We further quantified the differences between neuronal sub-populations by calculating the fraction of cells that were co-active in pairs of distinct task phases or periods (Fig. 6h for an example mouse; Supplementary Fig. 10 for all mice). Populations preferentially active in encoding and retrieval periods showed substantial overlap, whereas the subpopulation preferentially active during the maintenance period was clearly distinct. In some mice, however, the populations preferentially active during encoding and retrieval also clearly reflected preference for either right or left turns (Supplementary Figs. 9 and 10). In addition, within the encoding and retrieval periods, different sets of neurons preferred distinct task phases, indicating sequences of neuronal activation during both sample and choice runs.

Previous studies also found neural sequences specifically during the delay period of WM tasks in both mPFC[16] and dmStr[22]. Such sequences could help to maintain information in short-term memory. Therefore, we further evaluated the temporal order of the activity of mPFC→dmStr projection neurons during the maintenance period (phases 5–6 and 6–7). Some cells were consistently active at the beginning or at the end of the maintenance period whereas others displayed more variable and temporally distributed activity across trials (Supplementary Fig. 11). Overall, cells spanned the entire maintenance period with their activity (Fig. 7a, b for an example mouse; Supplementary Fig. 12 for all cells across mice). We tested for consistency and significance by splitting all correct trials into half, creating a 'training' and a 'test' set. We then ordered neurons according to the peak times of the mean signals across the training set and applied the same ordering to the 'test' set as well as to the mean ΔF/F signals of mistake trials. Whereas the temporal order of neuronal activation was similar in test and training data for correct trials, the sequence of activity deteriorated for mistake trials (Fig. 7b; Supplementary Fig. 12) with response peaks shifting significantly further apart from training data compared to test trials (Fig. 7c).

To further corroborate the notion of sequential activity, we also accounted for the observed trial-by-trial variability and evaluated whether neuronal pairs showed temporally ordered activation consistently across trials. To this end, we defined a novel 'binary directed orderability' (BDO) index that is a real number bounded between −1

and 1 ("Methods"). If a neuron $i$ is consistently active earlier than a second neuron $j$ in all trials, the neuron pair [$i,j$] is assigned a BDO of 1 (or −1 for [$j,i$] as BDO is antisymmetric) (Fig. 7d). Likewise, if the first neuron is active earlier than the second neuron in half of the trials but later in the other half, this pair is not orderable and is assigned a BDO value of zero. We computed BDO for all pairs of neurons, and sorted neurons by their average BDO (Fig. 7e for example mouse; for all mice see Supplementary Fig. 13). To quantify whether the overall order-ability observed exceeded that of shuffled data, we computed absolute value BDO averaged over all neuron pairs (ABDO), further averaged it over all mice, and compared it to shuffled data. We found that the average ABDO value across all neuronal pairs was higher compared to that of shuffled data in all periods of the task ($p < 0.001$), including the two phases of the maintenance period. (Fig. 7f). We also found that these orderability patterns can be fully explained by temporal preference of individual neurons to earlier or later parts of a given phase, as opposed to a more complex hypothesis of time-invariant order-ability, where individual cells would be orderable without having temporal preference (data not shown).Taken together, our results show that mPFC→dmStr projection neurons display orderable activity patterns in all task phases, including the delay period, and thus indicate the presence of neuronal sequences in this specific population not only during maze runs but also during WM maintenance.

Finally, we tested whether the activity of mPFC→dmStr projection neurons displays significant encoding of either choice behavior (left vs. right turn in choice runs) or performance outcome (correct vs. mistake; considering left and right sample runs separately; "Methods"). Across task periods, we found a relatively high fraction of neurons predictive for left vs. right turning in the encoding and retrieval periods but not in the maintenance period (Supplementary Fig. 14). This result corroborates the idea that for left vs. right turns partially distinct subpopulations are active (Fig. 6e and Supplementary Fig. 9). With respect to performance outcome, we found a high fraction of neurons predictive for outcome (correct vs. mistake) only during the late retrieval period (Supplementary Figs. 14 and 15). Taken together, our findings corroborate the notion that mPFC→dmStr projection neurons participate in maintaining the relevant information for spatial working memory.

## Discussion

Our study contributes to the long-standing goal of dissecting the brain circuits involved in different aspects of working memory. Whereas several key pathways have been identified and characterised, other pieces of the puzzle are still missing. For example, neurons in both dorsal and ventral hippocampus (HPC) interact with mPFC neurons in spatial WM tasks, with CA1 activity typically leading mPFC activity[44,45]. Hippocampal–prefrontal afferents engage particularly in the encoding phase of spatial WM and not during maintenance or retrieval[15]. The nucleus reuniens of the thalamus (RE) coordinates the coherent interactions across the mPFC–RE–HPC circuit[46,47]. Projections from mediodorsal thalamus (MD) to mPFC sustain prefrontal activity during the maintenance phase[8,39]. The reciprocal mPFC→MD pathway, in contrast, mainly supports subsequent choice and action selection[16,48]. Together, these results suggest that MD reinforces activity in mPFC neurons that encode task-relevant information essential for adaptive behavior[16,48].

Here, we complement previous work[2,3,8] on mPFC neuronal coding properties during spatial memory by demonstrating that mPFC→dmStr projection neurons engage during WM maintenance. Optogenetic inhibition of mPFC→dmStr activity reduced task performance specifically when applied during the maintenance period. Using two optical population recording techniques—without and with cellular resolution—we revealed that the mPFC→dmStr pathway shows higher activity during the late maintenance period in correct trials compared to mistake trials. Despite temporally sparse activity during

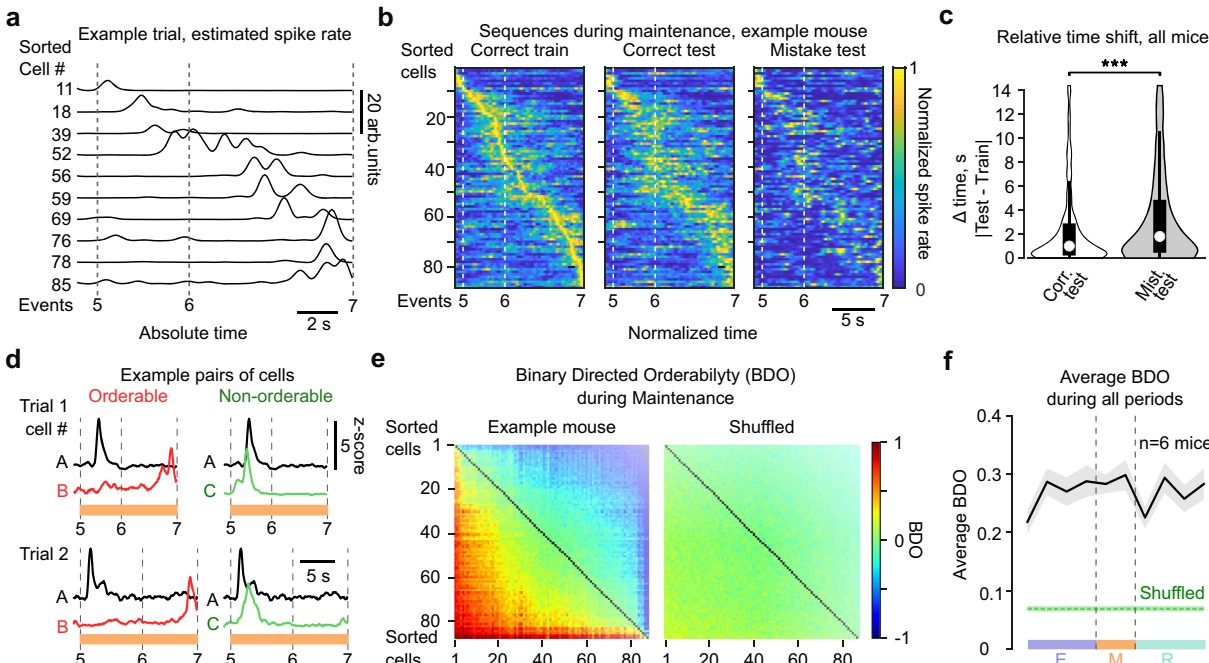

**Fig. 7 | Sequential activation of mPFC→dmStr neurons. a** Representative deconvolved ΔF/F traces from 10 example maintenance cells. Cell numbers correspond to the sorting in (**b**). **b** Normalized deconvolved neuronal signals during the maintenance period for an example mouse. Left: Mean signals for a randomly selected half of correct trials used as a training set. Middle: Mean signals for the remaining test set of correct trials. Right: Mean signals for mistake trials. In all plots, neurons are sorted according to the signal peak times in the training data. **c** Relative time shift of response peaks compared to training data for all neurons. Peak times shifted less for test trials (median = 1.0 s, 75th percentile = 2.8 s) compared to mistake trials (median = 1.80 s, 75th percentile = 4.8 s). ***$p = 6.62 \times 10^{-5}$, two-sided

Wilcoxon signed rank test, $n = 341$ neurons. **d** Examples of two trials, showing an orderable cell pair [A,B] (left) and a non-orderable pair [A,C] (right). Cell A is active earlier than cell B in most of the trials ($BDO = 0.96$), whereas cell C is active earlier than cell A in some trials but later in others, resulting in a lower $BDO$ of 0.66. **e** Left: $BDO$ matrix calculated for the maintenance period for all neurons recorded in an example mouse. Right: $BDO$ matrix for shuffled data. Neurons are sorted by their average $BDO$. **f** Average absolute value $BDO$ (±s.e.m.) across all neuronal pairs for six mice (black), calculated for all 10 task phases and compared to shuffled data (green). Source data are provided as a Source Data file.

the maintenance period and high trial-to-trial variability, neurons displayed significant temporal orderability and showed consistent preference for specific task phases in correct trials, leading to a sequence-like temporal activation pattern. During mistake trials, this temporal order of maintenance neurons at least partially fell apart. In agreement with this finding, mPFC→dmStr bulk signals—and to some degree the population activity patterns resolved at cellular resolution—encoded information about upcoming choice in the late maintenance period (correct vs. mistake choice). Such encoding was mainly evident for upcoming choice turns to the side contralateral to the recording site. In agreement with previous reports[8,16] we found weak evidence for spatial coding of mPFC neurons (right vs. left correct choice) during the maintenance period, whereas it was clearly present for encoding and retrieval periods. In contrast, other studies previously reported that a fraction of delay-related mPFC neurons is tuned to spatial information[3,24] and that the direction of the sample run could be decoded from the delay-period activity of mPFC neurons[23]. In yet another study, half of the recorded neurons in dmStr significantly encoded the sample stimulus at some point during the delay period[22]. Overall, the neural basis of how space and choice are encoded in mPFC microcircuits in alternation tasks remains unclear. Resolving this issue will require a more fine-grained mapping of mPFC areas and better dissection of specific subpopulations and their interactions with other brain regions.

Nevertheless, our findings suggest that mPFC neurons functionally organize downstream striatal activity throughout the entire WM maintenance period. Sustained mPFC activity during the maintenance period may create a ramping-up activity profile in dmStr, bridging the phase of short-term memory encoding to the subsequent action

(choice) similar to the situation described for evidence accumulation tasks[49]. This notion is in line with previous research demonstrating that persistent input from mPFC to dmStr represents decision variables[50] and controls temporal processing in dmStr[51]. Because neural sequences also have been found in striatum for tasks not involving spatial WM[52] and because sequential delay activity in dmStr was dissociated from stimulus encoding activity[22], delay-period spanning neural sequences in dmStr may have a more general role in preparatory activity for future choices rather than serving solely working memory.

Striatal activity correlates with successful execution of goal-directed actions[33] and our data support the idea that mPFC→dmStr activity contributes to correct action initiation following the WM maintenance period. Growing evidence indicates that mPFC and dmStr jointly contribute to successful execution of WM tasks[21]. For example, optogenetic activation of adenosine $A_{2A}$ receptors in dmStr selectively impairs WM performance when applied during maintenance and retrieval but not during encoding[17]. Clearly, dmStr is not the only downstream region that receives information from mPFC during WM maintenance. Other potential target regions include VTA[53] and the posterior parietal cortex (PPC)[54,55]. Furthermore, mPFC→dmStr neurons may have collaterals to other brain regions including contralateral PrL, ipsilateral insular cortex, and other cortical and subcortical regions[19,56,57]. The specific roles of these pathways during the distinct WM periods warrant further investigation. A full picture of the dynamic interactions within the hippocampal-prefrontal-striatal-thalamic loops and beyond has yet to be established. Our newly devised multi-fiber photometry approach, which enables simultaneous photometric recordings from several tens of brain regions[58], could be particularly well suited to measure large-scale signal flow during WM phases across

multiple regions interacting with mPFC. Multi-regional recordings could elucidate how coordinated activity in distributed circuits involving mPFC and multiple striatal regions may in general govern reward-related learning of goal-directed behaviors[59,60].

In addition to planning choice-related actions, prefrontal-striatal projections could be involved in impulsivity control, which is crucial for the successful execution of a WM task. In line with this notion, mPFC→dmStr projection neurons recently were found to selectively engage in inhibitory control[25]. In our experiments, perturbation of brain dynamics with the NMDA receptor blocker MK-801 induced hyperactivity, possibly by affecting such inhibitory control function of the mPFC→dmStr pathway. Consistently, pharmacologically induced impulsive behavior was partially restored and WM performance partially rescued by optogenetic enhancement of the mPFC→dmStr pathway. Indeed, MK-801 application was previously shown to increase decorrelated activity in mPFC neurons and to decrease organized burst activity induced by coordinated synaptic inputs[61,62]. We hypothesize that the extra 20-Hz phasic mPFC→dmStr pathway activation may have helped to restore organized cortical input to the downstream striatal circuits and thereby promoted correct action choices.

WM impairment in schizophrenia patients has been associated with the dysregulation of HCN channels, which are therefore increasingly recognized as an important therapeutic target for controlling WM dysfunction in neuropsychiatric disorders[63]. Here we investigated the modulatory function of HCN1 channels during the WM task. Consistent with their expression pattern[37] and complementing previous research[38,39], we demonstrated that blockade of HCN channels enhanced WM-related activity in mPFC→dmStr projection neurons. However, we did not observe improved performance in the spatial WM task, which may be due to the high-performance level of healthy mice. We reasoned that a behavioral effect might become obvious under conditions of WM impairment. Application of MK-801 induces a robust WM impairment[29,30] and can be considered to provide a relevant schizophrenia symptom animal model by blocking NMDA/glutamatergic signaling[31]. Indeed, co-administration of the HCN-channel blocker J&J12e partially rescued WM impairment induced by MK-801. A possible mechanism for this improvement could be an increased glutamate release probability, enhancing AMPA receptor-mediated, but not NMDA receptor-mediated, synaptic transmission, as was found during the co-application of the HCN-channel blocker ZD7288 and MK-801 on hippocampal slices[64]. Our results further motivate the use of HCN-channel blockers to alleviate disease conditions that impair WM. Further studies are necessary to better understand the synaptic and circuit[65] in vivo mechanisms of how these blockers affect working memory functions. Focusing on the function of HCN channels specifically in prefrontal-striatal projection neurons might be a promising avenue.

## Methods

### Animals

Experiments were performed on 34 male C57BL/6 mice and 4 male transgenic mice Ai148(TIT2L-GC6f-ICL-tTA2)-D (Nr. 030328, The Jackson Laboratory) for the miniscope experiments, all aged 6–8 weeks at the first use. Animal housing was organized by the Laboratory Animal Services Center (LASC) of the University of Zurich (www.lasc.uzh.ch). Mice with chronic implants were housed individually under a reversed 12-h light–dark cycle with food and water available *ad libitum* before behavioral training. During behavioral training, mice were water-restricted and maintained at 85% of their initial body weight. All experiments were performed during the animals' dark period. Experimental procedures were conducted in accordance with the guidelines from the Veterinary Office of Switzerland and were approved by the Zurich Cantonal Veterinary Office. We used only male mice for practical reasons and because only one T-maze setup was available. We do not expect sex differences regarding the investigated

WM mechanisms, which however will need to be tested in a separate study.

### Surgical procedures

Mice were first anesthetized with isoflurane (1.5–2%) and head-fixed in a stereotactic apparatus. The viral injections and fiber implantations were performed in the same surgery. First, after the small craniotomy, retrograde virus AAV-retro/2-hSyn1-chI-mCherry_2A_NLS_iCre (210 nl, $\sim 1 \times 10^9$ vg µl$^{-1}$) was delivered unilaterally into the left dmStr (+1.05 AP, −1.50 ML, −2.05 DV), corresponding to the medial anterior part of dmStr[18]. On the same day, another viral injection was delivered into the left PrL (+1.8 AP, −0.6 ML, −1.3 DV): we used AAV2.9.Syn.Flex.GCaMP6m (210 nl, $\sim 1 \times 10^9$ vg µl$^{-1}$) for calcium recording experiments, AAV2.5-CAG-FLEX-ArchT-GFP (210 nl, $\sim 1 \times 10^9$ vg µl$^{-1}$) for inhibition experiments, and AAV2.2-EF1a-DIO-hChR2(E123T/T159C)-mCherry (210 nl, $\sim 1 \times 10^9$ vg µl$^{-1}$) for activation experiments. For fiber photometry and optogenetic experiments, mice were unilaterally implanted with a ferrule-coupled optical fiber (0.22 NA, 400-µm diameter) after viral injections in the same surgical session. The distal end of the inserted fiber implant was cut and polished at a 45° angle and the flat side positioned immediately dorsal to the PrL area and directed towards the brain's midline. Finally, the fiber implant was fixed in place by glueing it to the skull with dental cement. For the miniscope experiments we performed a second surgery 12-14 days after the virus injection. For insertion of the GRIN lens (Gradient Index Lens; GRIN-TECH, Jena, NEM-100-06-08-520-S-0.5p; 1-mm diameter, 80-µm working distance, 4.54-mm length), we implanted a custom-made stainless steel guide cannula (1.2-mm outer diameter) with a glass coverslip (0.125-mm thick BK7 glass, Electron Microscopy Science) glued to the bottom. After cleaning the skull from periosteum and tissue, we performed a circular craniotomy of 1.3-mm diameter, centered above the mPFC (1.9 mm anterior and 0.4 mm lateral to Bregma,). We aspirated 1.2 mm of brain tissue overlying the targeted PrL area to prevent increased intracranial pressure when inserting the cannula. The guide cannula was lowered 2.3 mm (from the skull surface) into the craniotomy and fixed with UV-curable glue (Loctite 4305). For secure attachment and durability of the implant, two anchor screws (18-8 S/S, Component Supply) were placed on the contralateral parietal and the interparietal plate. Next, either blue-light curable Scotchbond Universal (3 M ESPE) or Metabond (Parkell) were applied around the implant, the screws, and the exposed cranium. For additional stability and for attaching a small metal head bar to the implant, a layer of dental acrylic (Paladur) was added on top of the first layer. Mice received postsurgical analgesic and anti-inflammatory treatment for three days with buprenorphine (0.1 mg/kg BW, s.c. every six hours during daytime, and 0.01 mg/ml in drinking water during the night) and carprofen (4 mg/kg s.c., twice a day).

### Histology

At the end of each experiment, mice were transcardially perfused with 4% paraformaldehyde in phosphate buffer, pH 7.4. Fixed tissue was then sectioned (100 µM) using a vibratome and mounted on slides with Fluoromount (Dako). Direct fluorescence of GCaMP6m, GFP-tag of ArchT or mCherry tag of ChR2 was examined under a confocal microscope (Fluoview 1000; Olympus) to assess the extent of viral spread and the axon-terminal expression pattern.

### Behavioral setup and training

Behavioral experiments were performed using a data acquisition interface (USB-6008, National Instruments) and custom-written MATLAB software to control devices required for the task and to record trial-related neural activity and licking data. Behavior training for the recording experiments started ~2 weeks after the surgery (~5 weeks for inhibition and miniscope experiments). First, mice were given 2 days of habituation to the T-maze. Habituation sessions

consisted of 5–10 min of free exploration in the maze with all doors open and freely accessible water rewards. On the subsequent 2–5 days, mice underwent behavioral shaping, which consisted of 10 trials per day running to baited goal arms in alternating directions. A drop of water (10 μl) was used as a reward. Reward delivery was triggered by licking and controlled with a miniature rocker solenoid valve (0127; Buerkert). For the miniscope experiments, mice were also accustomed to carry the miniscope by habituating them to its weight in 20-min habituation sessions in the home cage. Once mice followed the sequence of the shaping procedure at a speed of 1 min per trial, they underwent further training in the T-maze delayed non-match-to-place (DNMTP) task until they reached the desired performance criterion (>60% correct trials on 2 consecutive days). In the training phase, mice in the choice run had to choose the T-maze arm (right or left) opposite to the arm they had visited in the sample run before the delay. Each trial of the session consisted of three periods: encoding, maintenance and retrieval. In the encoding period (sample run) one of two goal-arms was blocked by a door and the mouse was directed towards a water reward in the open arm. During this period, the animal must encode the location of the sample reward. In training and recording sessions, sample arms were pseudorandomized (not more than three consecutive runs to the same sample arm). In the maintenance period, mice returned to the start box and had to maintain the sample reward's location in their working memory for a delay period (5-s duration for fiber photometry and miniscope experiments; 10-s duration for optical and pharmacological perturbations). In the retrieval period (choice run), all doors were opened and the mouse had to select the previously closed goal-arm to receive a second reward. Inter-trial delay was always fixed at 10 s. Performance of mice was maintained under 80% by providing 2–3 training sessions per week.

## Pharmacological interventions

First, mice were trained in the T-maze DNMTP task to reach the desired performance criterion: 2 days >60%. Next, mice were accustomed to the stress of systemic i.p. injection using the vehicle (0.3% Tween 80 in 0.9% NaCl; 10 ml/kg injected volume) 30 min before the beginning of the experiment. When mice reached the performance criterion under the vehicle injection conditions, they underwent the test day experiments with i.p. injection of either MK-801 (0.1 mg/kg) or J&J12e (3 mg/kg) or both drugs given in combination, 30 min prior to the test experiment. A fresh compound solution was prepared on each test day in the vehicle solution (0.3% Tween 80 in 0.9% NaCl; 10 ml/kg injected volume), sonicated for 30 min, and well mixed. In the first experimental design (Fig. 4b, Supplementary Fig. 4a, b), we trained mice in the DNMTP task with a 5-s delay, followed by the baseline day with the same 5-s delay (after vehicle injection). Next, we tested the mice's performance with a challenging 10-s delay after systemic injection of either the J&J12e compound or vehicle (J&J12e or Vehicle probe days respectively). To balance cohorts of mice, 50% of mice underwent vehicle testing first, followed by testing with J&J12e, and the other 50% of mice started with J&J12e testing, followed by vehicle testing. In the second experimental design (Fig. 4c), the delay was kept at 10 s for all test days. To balance cohorts of mice, 50% of mice underwent the MK-801 test first, followed by testing with the J&J12e and MK-801 in combination; for the other 50% of mice, the test order was reversed. Following the first test day, mice were given a 5–6 day break in their home cage without experiments.

## Fiber photometry

Pathway-specific photometric recordings were carried out through the implanted 400-μm diameter optical fiber using 0.3 mW of 488-nm excitation light (OBIS LX 488-nm laser, Coherent), modulated at 490 Hz. The 425-nm light (LuxX 425 laser, Omicron) used to control for motion artefacts was modulated at 573 Hz and delivered together with the 488-nm laser light through the same fiber. The dichroic beam-splitter

(No. F58-486, AHF) directed excitation light into the objective (F240FC-532, Thorlabs) and transmitted fluorescence signals. The fluorescence from the GCaMP6 expressing neurons was propagating along the optical fiber towards the photometry setup, was collimated by the same objective (F240FC-532), passed the emission filter (525/50 nm, No. F37-516, AHF) and was focused by the condenser lens (ACL3026U-A, Thorlabs) onto a photomultiplier tube (PMT; H10770(P)A-40, Hamamatsu). The photocurrent was amplified in the custom-built pre-amplifier and transmitted to the data acquisition board (DAQ USB 6211, National Instruments). The fluorescence signals were recorded at 2 kHz continuously until the end of each session and demodulated with the digital lock-in detection (MATLAB R2019b). Fluorescence contributions excited by the 425-nm and 488-nm lasers were demixed by demodulation at the respective modulation frequencies in 50-ms time bins, resulting in an effective 20-Hz rate for the fluorescence recording. At the beginning of each recording session, the mouse was placed into the start box before the first trial. Mice were simultaneously video tracked with the IR illuminated camera at 15 Hz frame rate (DMK 33UX178, Imaging Source). All continuous behavioral parameters such as the licking response were also simultaneously recorded on the same data acquisition board (DAQ USB 6211, National Instruments).

We used a custom-written MATLAB program for simultaneous data acquisition and DNMTP T-maze task control. For synchronization, several parallel routines were triggered simultaneously: (1) PMT signal recording for fiber photometry (via USB-interface on NI DAQ 6211); (2) Video recording for mouse tracking; (3) An online video analysis routine, which used image intensity changes in specific regions of interest (ROIs) to detect when the animal reached key positions in the maze (start box; exit of start box; end of corridor before T-junction; entry positions in the two T-maze arms; reward zone). These online detected events were timestamped and, together with the lick sensor, they were used to control the opening/closing of the maze doors and the waterspout valve opening according to the task structure.

## Optogenetic manipulation

Pathway-specific optogenetic inhibition experiments were carried out using constant illumination with 20-mW, 561-nm light for activation of ArchT. For pathway-specific activation in ChR2-expressing mice we applied 20-Hz pulses of 10-mW, 473-nm light, delivered through a 400-μm diameter optical fiber (NA 0.22). In each session employing optogenetics, perturbation light was delivered in 50% of randomly interleaved trials and targeted exclusively to one of the task periods (e.g., only during encoding period for one session and only during maintenance in a different session). To account for the trial-to-trial variability, the start and end of laser illumination was controlled by the online detected mouse position related to the key events of the respective task periods. In total, each mouse experienced 9 sessions, with 3 perturbation sessions for each task period.

## In vivo electrophysiological recordings

We validated optogenetic excitation and silencing of mPFC→dmStr projection neurons by performing acute in vivo electrophysiology. Mice (n = 4) were anesthetized with isoflurane (2% for induction; 0.8% during recording) and body temperature was maintained at 37 °C using a heating pad. A small craniotomy (1-mm diameter) was performed to provide access to the left PrL and the brain was covered with silicon oil. A silver wire was placed in contact with the cerebrospinal fluid through a small (0.5 mm) trepanation over the cerebellum to serve as a reference electrode. A silicon probe (Atlas Neurotechnologies, 16 linear sites, 100-μm spacing) was inserted through the craniotomy into the left cortical hemisphere to record multi-unit activity from the injection site in the left PrL and surrounding cortex. After insertion of the silicon probe, we positioned an optical fiber (0.2-mm diameter) directly next to the probe, directed downwards towards the brain. We waited 30 min to allow the recording to stabilise after

implantation of the electrode array. After stabilisation, the broadband voltage was amplified and digitally sampled at a rate of 30 kHz using a commercial extracellular recording system (RHD2000, Intan Technologies). The raw voltage traces were filtered offline to separate the multi-unit activity (MUA; bandpass filter 0.46–6 kHz) using a fourth-order Butterworth filter. Subsequently, the high-pass data were thresholded at 5.5 times the standard deviation across the recording session and the numbers of spikes in windows of interest were counted. Once the recording was stable, we performed light stimulation through the optical fiber to drive the expressed opsin. We delivered light by connecting the fiber-optic cannula via a fiber-optic patch cable to a laser (either OBIS 473 nm LX for ChR2, or OBIS 561 nm LS for ArchT). The CW laser was gated in a stepwise manner with a function generator, triggered by a TTL-pulse from the electrophysiology computer. We found cells directly driven by the laser stimulation, as well as cells presumably showing secondary effects via connectivity with driven cells. To combine data across mice for ArchT experiments, MUA activity at sites with clear light-induced responses was expressed as percent change from the average spike rate during the baseline 10-s pre-laser stimulation period.

### Behavioral analysis

All behavioral analysis was performed using custom-written MATLAB scripts. First, based on the behavior videos and licking actions, we identified 11 salient events in each of the 1085 trials collected from six mice. For correct trials the events were the following (Fig. 1a): 1–start of the sample run, 2–turning at the T-junction of the main maze arm, 3–first water reward, 4–end of licking period, 5–turn to run towards the start box, 6–reaching the start box, 7–start of the choice run, 8–turning at the T-junction, 9–second water reward, 10–end of licking period, 11–end of choice run. Because mice did not receive a reward in incorrect trials, events 9 and 10 were omitted for these trials. These 11 salient trial-related events defined 10 trial phases and were used for the analysis of optical recordings and video analysis of mouse behavior. The three major task periods correspond to phases 1–5 (encoding), 5–7 (maintenance), 7–11 (retrieval).

To address potential concerns whether any behavioral motor variable during the maintenance period could predict future choices (turns) of mice, we tracked mice with DeepLabCut[66] (DLC). In every frame we estimated coordinates of the mouse's nose, ears, and the tail-base. Next, from the coordinates of the mouse's nose, ears, and tail base (estimated in each video frame), we calculated geometrical and behavioral parameters potentially relevant for the T-maze DNMTP task: (1) the angle of mouse orientation (nose-to-tail base) relative to the axis of the T-maze main corridor (from start-box to T-junction); (2) the number of turns during the maintenance period, with a turn defined as changing head direction from facing the T-junction to facing opposite or vice versa; we separately counted clockwise and counterclockwise turns. In addition, we estimated (3) the position in the maze, (4) the movement (estimated as the frame-to-frame change of the nose coordinates divided by the time between frames); when mice moved along the maze this variable represented speed, and (5) the time of the first re-orienting turn towards the door after arrival in the start-box. Potentially, all these behavioral variables could carry some explanatory power of the future behavioral choice and/or the measured prefrontal calcium signals.

For the dimensionality reduction of all behavioral motor variables, we used UMAP embedding with the MATLAB library from Connor Meehan, Jonathan Ebrahimian, Wayne Moore, and Stephen Meehan (Uniform Manifold Approximation and Projection (UMAP), 2022, https://www.mathworks.com/matlabcentral/fileexchange/71902).

### Analysis of photometric calcium signals

To visualize pathway-specific activity, the recorded fluorescence signals were expressed as percentage change in fluorescence $\Delta F/F = (F(t)-$ $F_0)/F_0$, where $F(t)$ denotes the fluorescence value at time t across the entire trial time (from event 1 to event 11) and $F_0$ the baseline fluorescence level calculated as the mean value in the 1-s time window before the maintenance period (before event 5). For analysis across trials and mice, we z-scored the fluorescence signals for individual trials by calculating $(F(t)-F_0)/\sigma$, where $\sigma$ is the standard deviation of the fluorescence values in the 1-s baseline window for $F_0$ calculation. Because mice behaved freely in the T-maze, trial phases varied in their duration. To temporally align trial-related signals we therefore resampled the z-scored signals for each trial phase, as defined above, to match the median duration of this phase across all trials. The median duration was used for plotting. This registration of the recorded signals by segment-wise temporal resampling permitted us to average and compare the activity patterns in the separate task phases across trials and mice. All mice contributed 4 expert sessions to the dataset, which in total comprised 1085 trials (544 rightward sample runs: 407 correct, 137 mistakes; 541 leftward sample runs: 405 correct, 136 mistake). The resampled traces were used for detailed analysis of the maintenance period (between events 5 and 7) in Figs. 2 and 4.

To control for potential motion artefacts and hemodynamic signal components in our fiber photometry data of mPFC→dmStr pathway activity, we used two experimental strategies: First, we recorded task-related photometric fluorescence signals in a separate cohort of GFP-expressing control mice (4 mice, 5 sessions each). Second, we performed simultaneous GCaMP6m measurements at two excitation wavelengths (488-nm and 425-nm excitation for calcium-dependent and calcium-independent fluorescence, respectively). To this end, both lasers were modulated at different carrier frequencies and the fluorescence signals were digitally demodulated (see above). The 425-nm excited control fluorescence traces were clearly smaller than the signals observed with 488-nm laser excitation and relatively flat (Supplementary Fig. 2a, b). Control photometric signals from GFP mice were also small and relatively flat, with some negative-going fluorescence dips appearing for trial phases 3–4 and as mice approached the waterspout and licked to collect the reward. In summary, these control experiments exclude hemodynamic or motion-related artefacts as major confounds and indicate that the maintenance signals are of neuronal origin.

### Miniscope imaging and data analysis

Four weeks after GRIN lens implantations, mice with GCaMP expression were taken into behavioral experiments. We used a head-mounted miniaturized microscope (nVista, Inscopix) for calcium imaging with cellular resolution. All mice were recorded at 20 Hz with a gain of 3–4 and 10–25% LED illumination (0.2–0.5 mW/mm²). Simultaneously, we video tracked the mouse and recorded behavioral parameters such as water reward delivery and licking response. After mice completed several expert sessions, calcium imaging data were pre-processed using Inscopix Data Processing Software (version 1.2.1), allowing for motion correction and extraction of cellular calcium signal traces. For efficient data processing, movies were spatially downsampled by a factor of 4 and temporally downsampled by a factor of 2. After applying motion correction in the Inscopix software, $\Delta F/F$ was calculated pixelwise using the mean fluorescence across all frames as $F_0$. Then, we applied the Inscopix PCA-ICA algorithm to automatically identify the spatial location of cells and their activity profile throughout the recorded session. Finally, in order to match cells recorded over several sessions, we employed the longitudinal registration algorithm. For further analysis, we used only those cells that were successfully recorded longitudinally in all expert sessions. For comparison with the fiber-photometry data, we summed the pre-processed activity of all longitudinally tracked neurons (28–88 neurons per mouse) for each mouse ($n = 6$ mice, 3–5 sessions per mouse) and treated these population signals in the same way as the bulk fluorescence signals measured by fiber photometry.

## Analysis of single-cell activity across task periods and task phases

We evaluated the degree to which neurons were specifically active in defined time windows (either the 3 task periods or the 10 task phases defined as the intervals between the 11 salient T-maze events). To optimize temporal accuracy and reduce the amount of non-spike fluctuations in the signal, we temporally deconvolved the pre-processed miniscope calcium imaging data using a supervised algorithm based on deep neural networks[43]. Next, we calculated for all neurons and each trial the mean activity in each time window considered (either periods or phases). Based on a rank sum test of mean activity values for each time window across all correct trials against the pool of all other time windows, we identified the significance level of a given neuron being active in a particular time window. The significance level is presented as $-\log_{10}(p)$, where $p$ is the $p$-value of the rank sum test (e.g., if $p = 0.01$, $-\log_{10}(p) = 2$). To quantify the fraction of neurons significantly active in specific time windows (or pairs of time windows), we binarized the significance levels by thresholding at $p = 0.01$. For either task periods or task phases, we plotted these neuronal fractions as a symmetric matrix, with values on the diagonal representing the fraction of neurons displaying significant activity in at least this time window, and off-diagonals showing the fractions of neurons active in both of the respective time windows.

## Neuronal encoding of behavior

To analyze encoding of behavior by the bulk activity of mPFC→dmStr projection neurons (based either on the photometry data or the summated cell activities for miniscope imaging) we used receiver operating characteristics (ROC) analysis. We considered three behavioral aspects: left vs. right turning direction and correct vs. mistake performance in choice runs, considered separately for left and right sample runs. We used the z-scored calcium signals for all expert sessions (>60% correct trials on 2 consecutive days), resampled to align them to the task phases. We also applied ROC analysis to the experimental data with pharmacological intervention (J&J12e correct vs. mistake; and J&J12e vs. vehicle; Fig. 4 and Supplementary Fig. 4). For each pair of trial types considered, we randomly selected 100 trials for each group (20 times repeated) and calculated the area-under-the-curve (AUC) for the ROC curve. We performed this analysis for each time bin across the full time window of interest using a 3-point moving window. To test for significance of classification of the two trial types, we additionally performed the same analysis for data with shuffled trial labels. In the figure plots we show the mean ± s.e.m. of the AUC values for real and shuffled data (for each time point and considering the 20 draws). We used a Mann–Whitney $U$ test ($p < 0.01$) to evaluate the difference between real and shuffled datasets and test whether trial type classification was significant.

To evaluate neuronal encoding on the population level, based on the miniscope data, we tested for each neuron whether in a given time window (task period or task phase) its mean activity was predictive of behavior (either for left/right turning direction or for correct/mistake performance in choice runs; ranksum test, $p < 0.01$). For each time window, we then counted the number of predictive neurons and tested if this number was significant by using a binomial model, in which each neuron, as null hypothesis, could independently be significant with probability $p = 0.01$. For all mice, we plotted the fraction of cells whose activity was significantly predictive of the behavioral classification considered, e.g., left vs. right turns, for task periods and for task phases in Supplementary Fig. 13a and b, respectively.

Finally, we investigated if specific behavioral aspects were encoded transiently during the maintenance period. As for the above analysis, we focused on left vs. right turning behavior and also analyzed correct vs. mistake performance, separately for left and right sample turns. We computed the euclidean (L2) norm between neuronal population signals, using the deconvolved ΔF/F traces averaged for each of the two behavioral conditions, and tested the result against a null model with condition labels shuffled ($p < 0.01$, permutation test). We performed this test for each time bin of resampled trial time and plotted the significance level as $-\log10(p)$ as a function of the resampled time (Supplementary Fig. 14). We also performed transient decoding using cross-validated machine learning methods: linear regression, logistic regression and two-layer neural network (data not shown). Results were largely consistent with L2 norm for direction discrimination, but showed no effect for performance discrimination. We concluded that these methods do not converge to optimal performance due to a low amount of mistake trials.

## Orderability analysis

A simple pairwise correlation metric was not sufficiently precise to quantify the activation sequences in neuronal populations under conditions of multiple peaks, variable trial durations, and noise. Therefore, we introduced the 'Binary Directed Orderability' (BDO) index as a more robust method. For each pair of neurons ($i, j$) and each trial, BDO evaluates whether most of the signal of neuron $i$ came before that of neuron $j$ or vice-versa (hence the name binary), then finds the average frequency over trials of the neuron $i$ being earlier than neuron $j$. Thus, this metric determines whether neuron $i$ is consistently active earlier than neuron $j$ across trials, independent of the magnitude of the time differences and of the exact time period in which neurons are active. For each active neuron, we first calculated the probability $p(t)$ of being active at time $t$ by normalizing the baseline-subtracted deconvolved signal $x(t)$ (after baseline subtraction) so that the sum over all time bins $1..N$ is one:

$$p(t) = \frac{x(t)}{\sum_{t=1}^{N} x(t)} \tag{1}$$

Next, we determined the center of mass μ of $p(t)$, i.e., the time bin in which the neuron is most active:

$$\mu(p) = \sum_{t=1}^{N} t\, p(t) \tag{2}$$

We prefer to use μ rather than the peak time because μ is more robust against small perturbations when calculated for each noisy trial and corresponds to the weighted average time of the entire neuron trace, not of one local event. For each pair of neurons [$i, j$] and each trial $k$, we defined the binary order $T_{k,ij}$ as one if neuron $j$ was active later than neuron $i$ ($\mu_j > \mu_i$) and as zero otherwise. Thus, the fraction of trials, in which neuron $j$ was active later than neuron $i$, is given by

$$f_{ij} = \frac{1}{N_{trials}} \sum_{k=1}^{N_{trials}} T_{k,ij} \tag{3}$$

where $N_{trials}$ is the total number of trials. Because the case $\mu_j = \mu_i$ can be neglected for floating point comparison, it follows that $f_{ij} + f_{ji} = 1$.

Finally, the BDO index is defined as the linear function of the above fraction

$$BDO_{ij} = 2f_{ij} - 1 \tag{4}$$

The BDO index is bound between [−1, 1]. It is zero when no consistent order occurs across trials, +1 if neuron $j$ consistently follows neuron $i$, and −1 if neuron $j$ leads neuron $i$. For visualization, we plotted the BDO for all neuron pairs as a heatmap, where the hue denotes BDO value. For validation, we compared this plot to a shuffled heatmap, for which a shuffle over neuron labels was performed for each trial individually. To quantify whether BDO was above chance, we defined the average absolute value BDO (ABDO) as the mean absolute value of the

off-diagonal *BDO* values:

$$ABDO = \frac{1}{N_{neuron}^2 - N_{neuron}} \sum_{i \neq j} |BDO_{ij}| \qquad (5)$$

where $N_{neuron}$ denotes the total number of neurons. If *ABDO* is significantly larger than for the shuffled heatmap ($p \leq 0.01$, permutation test), this means that the signals recorded from neurons in this population are on average better pairwise orderable than random signals.

## Statistics and reproducibility

For all mice which entered the dataset we confirmed mPFC targeting using histological analysis similar to the examples shown for photometry (Fig. 1d), optogenetic perturbations (Figs. 3a, 5a), and miniscope analysis (Fig. 6a, b). In all time-dependent plots with shaded error bar (e.g., Fig. 2a) the solid line shows the mean value at each time point and the shaded error bar s.e.m.. All bar plots represent mean values with s.e.m. error bars. In experiments where the pairs of observations were made during the same behavioral session, such as Light Off vs Light On conditions in optogenetic experiments, we used two-sided Wilcoxon signed rank test. In experiments with separate observations, such as comparing ΔF/F levels measured during different sessions (e.g., in pharmacological experiments of Fig. 4a) we used Wilcoxon rank sum test (also called Mann–Whitney *U*-test). All *p*-values were corrected either with a Tukey or Bonferroni post hoc test to the number of test groups. If, after Bonferroni correction, *p*-values were exceeding 1, they were set to 1. Miniscope experiments with cellular resolution enabled observations of many neurons within each mouse; therefore for each period of the task we could construct a distribution and use the binomial test to evaluate whether there were more neurons preferentially active in one phase of the trial compared to another phase, with the null hypothesis being that it is equally likely to find neuronal preference for each period of the trial; *p*-values were pooled across mice and Fisher's method used to confirm significance across all mice (Fig. 6f). Source data are provided for every figure as a Source Data file. Calculated *p*-values are represented in figure panels as follows: *$p < 0.05$; **$p < 0.01$; ***$p < 0.001$; ns, not significant.

## Reporting summary

Further information on research design is available in the Nature Portfolio Reporting Summary linked to this article.

## Data availability

Data from photometry recordings, miniscope recordings, and behavioral video tracking that were generated in this study have been deposited in the Zenodo database under accession code https://zenodo.org/doi/10.5281/zenodo.8387631. The processed data of task performance for optogenetic perturbation experiments are provided as source data. For each figure, Source data are provided with this paper.

## Code availability

Custom-written code in MATLAB (R2020b) and Python 3 has been deposited in the Github repository (https://github.com/HelmchenLabSoftware/wilhelm-sych-tmaze-analysis).

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

## Acknowledgements

We thank Stefan Giger, Hansjörg Kasper and Martin Wieckhorst for technical assistance, Lazar Sumanovski for assistance in brain histology, Ariel Gilad and Peter Rupprecht for advice on data analysis. We thank Yasir Gallero-Salas for helpful discussions and Philipp Bethge for comments on the manuscript. This work was supported by a Transfer Project and an IPhD Project from SystemsX.ch (51TP-0_145729 and 51PHPO_157359; to F.H.), and through a Roche Joint Collaborative Project. Further support came from the European Research Council (ERC Advanced Grant BRAINCOMPATH, project 670757 to F.H.), the Swiss National Science Foundation (310030_192617 to F.H.; CRSII5-173721 and 315230_189251 to B.F.G.), the German-Swiss Research Unit "Barrel Cortex Function" (DFG FOR1341, SNSF 310030E-147485; to F.H.), the Human Frontiers Science Program (RGY0072/2019 to B.F.G), the ETH Zurich (ETH-2019-01 to B.F.G.), and the University of Zurich (Forschungskredit to C.L.). B.F.G. and F.H. acknowledge support by the University Research Priority Program (URPP) 'Adaptive Brain Circuits in Development and Learning' (AdaBD) of the University of Zurich. This work was also supported by CNRS (contract UPR3212), the FRM ("Amorçage de Jeunes Equipes" AJE202110014579 to Y.S.) and by ITI NeuroStra (Y.S.) from the University of Strasbourg.

## Author contributions

M.W. and F.H. conceived the project and designed all experiments. M.W. and Y.S. carried out experiments. C.L. performed and analyzed multi-unit electrophysiology recordings for all optogenetic experiments. L.S.C. assisted in fiber-optic related surgeries and experiments. J.L.A.W. automated the T-maze DNMTP task. M.W., Y.S., and F.H. analyzed the fiber photometry data. R.B. and E.A.A. performed the surgeries and assisted in the miniscope experiments. Y.S. and A.F. contributed to photometry analysis and performed the majority of the analysis for miniscope experiments. B.J.H., E.C.O., Y.S., A.F., and B.G., assisted in the study design and interpretation of experimental results. M.W., Y.S., and F.H. wrote the paper with comments from all the authors.

## Competing interests

The authors declare no competing interests.
