## [Peer Review File · Nature Communications]

Striatum-projecting prefrontal cortex neurons support working memory maintenanceREVIEWER COMMENTS

Reviewer #1 (Remarks to the Author):

NC review 2021-12-14

Using fiber photometry and miniscope imaging, Chernysheva and colleagues reported three sets of experimental data in support of the main conclusion that striatum-projecting prefrontal cortex neurons support working memory maintenance. (1) there was strongest neural activity during the delay (maintenance) phase of WM in mPFC→dmStr projection neurons; (2) Optogenetic inhibition of mPFC→dmStr neurons selectively during the delay phase impaired performance while optogenetic activation of these labeled neurons during the delay alleviated WM impairment induced by MK-801; (3) miniscope imaging revealed the increased fraction of neurons (but not amplitude) during WM maintenance with apparently sequential activation patterns. The experiments were carefully designed and skillfully executed and the data analysis was sophisticated and apparently sound. These findings extend the previous studies and reinforce the conclusion, albeit not entirely novel, on the critical role of mPFC neurons in the working memory maintenance by isolating these mPFC→dmStr projection neurons and on sequential firing of mPFC→dmStr neurons to support WM maintenance.

However, there are several major concerns that need to be addressed before the manuscript to be considered for publication in NC.

1. Because ChR2 activation during the maintenance period produced only a small increase (about 3%) in WM performance, the authors suspected that the high (~80%) baseline performance may have masked ChR2-mediated potential improvement. To address this issue, they used MK801 to impair WM (below 70%) and then showed that ChR2 activation during the delay (but not the sample and choice) phase increased WM performance. However, the baseline performance ("Off") in these three phases were quite different (~60% sample, 45% delay and 70% in choice). The preferential improvement by ChR2 in the delay phase may be confounded by its particularly "low" baseline. The authors need to carefully address this issue to support the conclusion.
2. The strongest increase in calcium signaling is at the event 5, the turning of T maze. This calcium signal increase can have alternative interpretation, i.e. calcium signal response to position cue (i.e the turning point). Such stimulus-trigger increase in calcium signal in response to position cue has been shown in the recent recording of calcium in astrocytes and neurons (see Lin et al. *Entrainment of Astrocytic and Neuronal Ca²⁺ Population Dynamics During Information Processing of Working Memory in Mice*. *Neurosci Bull.* 2021 Oct 26). This needs to be carefully addressed. Furthermore, the standard definition of the delay phase starts at the close of the rest chamber (event 6). There was no further calcium increase in event 6 (compared to event 5). Therefore, the main increase in calcium signaling seem to occur before the maintenance. This requires careful evaluation and interpretation. It is important to dissect out the contribution of "position cue" to the calcium increase in WM information processing.
3. It is important to provide the calcium signal data after MK801 and HCN-channel blocker J&J12e, so that possible correlation of these calcium signal can be made with their effects on WM performance.
4. While fiber photometry recording of calcium signaling neural populations show clear increase in the maintenance phase (5-7), miniscope of individual neurons show only a minor increase in the fraction of neurons that fired above the threshold during this phase. However their mean spike activity was in fact lower than that of other two phases. This needs to be acknowledged and carefully interpreted.

Minor points:

The study by Liu and colleagues "Medial prefrontal activity during delay period contributes to learning of a working memory task, *Science*. 2014 Oct 24;346(6208):458-63" should be cited in the Introduction.

Please provide more detailed descriptions of coupled analysis of event-specific calcium signaling with behavioral parameters (events 1-10) including apparatus, label, data analysis.

Reviewer #2 (Remarks to the Author):

Chernysheva et al. studied striatum-projecting prefrontal neurons in mice during a spatial-alternation task. The data are well presented and the manuscript well-written. The experiments will be of interest to those studying mPFC->dmStr during behavior. Limitations in the behavioral task design and missing control experiments weaken some of the conclusions. Overall, it's unclear how the present results substantially advance understanding of working memory and the mPFC->dmStr pathway.

1. The behavioral task design does not allow for strong conclusions. First, it's not clear different mice used similar behavioral strategies. Training occurred over a few days, and the variance in behavior is very large (Fig. 3d, Fig. S1a, Fig. 4b,c). Behavioral effects of mPFC silencing (9% performance decrease in Fig. 3d) are much smaller than differences across mice. At the very least, there should be a more thorough description of the behavioral data (e.g., showing learning curves, correlating them to effects of mPFC silencing).

Second, referring to the task as requiring working memory (WM) is an assumption not supported by the data. Mice could conceivably solve the task without WM (e.g., prepare a policy of left-right-left-right... in advance). I suggest revising the language to reflect this (e.g., "delayed alternation" instead of WM), although I appreciate that this is a term of art in the literature for this behavioral task.

2. The relationships shown in Fig. 2d have very small R^2 values and thus should be interpreted with much more caution. It looks like the three rightmost points in each scatter plot contribute a lot of leverage to the regression. Does it survive removal of those points? Similarly, the interpretation that RT decreases with change in late-delay-phase activity implies higher decision confidence is dubious. RT and confidence are often correlated, but needn't be. This latter statement could be better supported with references.

3. There are several relevant papers that would be useful to discuss in the context of the present results, including prior demonstrations of sequential activation in similar tasks in rodents (mostly rats):

- a. Brito et al., *Exp Brain Res*, 46, 52, 1982
- b. Jung et al., *Cereb Cortex*, 8, 437, 1998
- c. Baeg et al., *Neuron*, 40, 177, 2003
- d. Fujisawa et al., *Nat Neurosci*, 11, 823, 2008
- e. Horst & Laubach, *Neuroscience*, 164, 444, 2009
- f. Horst & Laubach, *J Neurophysiol*, 108, 3276, 2009

4. Were there spatial relationships between calcium dynamics of single neurons within a field of view? Others have observed layer-specific activity in this region. The imaging geometry here may allow for a laminar analysis.

5. The distributions of durations of encoding, maintenance, and retrieval for correct choices and mistakes are interesting. The means are stated to be no different from each other, but it looks like the variance of the "encoding" phase is larger on mistake trials versus correct trials. Is that true? It also looks like there is a trend for the mistake trials to have shorter durations in the "retrieval" phase. Do these observations impact the interpretation of the neuronal data?

6. The conclusion that "...performance decrease was not simply induced by altered locomotion" (line 203) does not completely follow from the observation that mean duration of choice runs were similar (Fig. 3e). For example, different movement kinematics produced by bouts of increases and decreases in velocity would produce the same choice run duration as constant velocity. Particularly in a task in which movement preparation can be used to perform (e.g., without WM), this claim and the manuscript overall would benefit from a more thorough analysis of movements. The authors allude to this somewhat in the statement on lines 175-177, when analyzing ipsi- versus contralateral movements (Fig. S2).

7. The effects of HCN blockade appear confined to the Maintenance period in Fig. 4a, but look

more nonspecific in Fig. S4a (i.e., a DC shift for drug relative to vehicle). Please clarify this, and the degree to which these signals can be compared appropriately across sessions.

8. The effects in Fig. 4c are interesting and appear bimodal. That is, it appears that mice that started off worse at the task had little-to-no effect of MK-801+J&J12e, whereas those that had higher initial performance had a substantial drop in performance with the combination of drugs (apparently not as bad as with MK-801 alone). First, is this true? If so, does it affect the interpretation (e.g., there could be two strategies of solving the task, one involving WM and one not)? Second, please clarify whether the statistical tests in Fig. 4c included omnibus test (e.g., ANOVA followed by planned contrasts).

9. The ArchT inactivation experiment (Fig. 3) requires a fluorophore-alone control.

10. I'm unclear about whether the BDO shuffle control is appropriate. Depending on the sampling distribution of $\langle \text{BDO} \rangle$, using a permutation test to compare to shuffled data could come with dangerous assumptions (why not bootstrap?). In any case, wouldn't simply cross-correlating $x(t)$ for neurons i and j , with the data split in half for cross-validation, be appropriate here?

Minor comments

1. The sentence on lines 50-52 could be qualified by a statement such as "...major axonal projections include..." or a reference to species differences. For example, mPFC across mammals projects strongly to other neocortical areas, hippocampus, and other midbrain structures than VTA.

2. I'm not sure I see how the experiments test "the computational role of the mPFC->dmStr activity" (lines 72-73).

3. What produced the decreased GFP signal in Fig. 2a? Does this impact the interpretation of the decreased GCaMP signal during those two time periods?

4. Please show the data underlying the means and SEMs in Fig. 3c (e.g., scatter plots, or line plots underneath, as in Fig. 3e).

5. The "Photometry analysis" section in the Methods would benefit from a more thorough description of control analysis (i.e., GFP channel).

6. The very large diameter implants (0.4 mm or 1.2 mm cylinders) clearly caused substantial damage to the areas dorsal to PrL. Does this affect the interpretation of the data?

Reviewer #3 (Remarks to the Author):

The authors have studied the role of striatal projecting prefrontal neurons (mPFC to dmStr) in guiding working memory behavior. Their study employs pathway-specific fiber photometry, optogenetics, pharmacology, and miniscope calcium imaging in mice performing a T maze alteration task. They find that neural activity is greatest during the maintenance period of the task, in which mice must maintain a working memory of the correct choice. They also find that perturbing neural activity selectively during this maintenance period produces the greatest change in behavior. From these results the authors conclude that mPFC to dmStr projecting neurons play an important role in maintenance of working memory. The results are noteworthy for shedding light on the role of corticostriatal neurons in regulating working memory. Furthermore, the result that these neurons appear to play the most important role during the maintenance period is novel and likely to interest the field. However, the lack of detailed behavioral characterization in the WM task made it difficult to fully accept the results as presented without ruling out other possible interpretations. The manuscript is clearly written for the most part, and this work presents data which have the potential to significantly interest the field, but the reviewer felt that some crucial

details to rule out potential flaws were missing. Furthermore, it wasn't clear how the sequential activity analysis fit in with the rest of the study.

Major comments:

1. The photometry results in Fig. 2 are interesting, but it's important to get some context about what the mice are actually doing during the maintenance phase of the task. Specifically, it seems important to know whether, during the delay period, mice sit still, or tend to rotate in the direction of their intended choice. For example, if they have to make a left choice, do they tend to show more leftward rotations during the delay period on correct versus incorrect choice trials? If so, this would present a potential confound in the photometry and miniscope results. Similarly, the optogenetic inhibition data in Fig. 3 are interesting, but it's important to know whether the laser has any effect on the animal's motion during the maintenance phase itself. For example, if mice tend to rotate in the direction of their intended choice, but the laser impairs this rotation, that presents a potential confound. This concern would largely be addressed if the mice just sit still or are generally not very active during the maintenance period.

2. Fig. 3d is a very important and potential interesting result, but several concerns were noted. First, the authors did not provide exact p values for all plots. Second, on line 200 they state $p = 0.0035$ yet in the figure legend they show one star for significance ($* p < 0.05$). This needs to be clarified. Third, the source of high variability of the data in the middle panel (light on during maintenance) compared to the left and right panels was unclear and a bit concerning. This led the reviewer to wonder whether the authors are not capturing some important feature of animal behavior that is taking place during the maintenance phase. Finally, it appears the authors are pooling across the ipsilateral and contralateral turn trials, but could they separately check whether performance is differentially affected on ipsi vs contra trials?

3. The sequential activity analysis in Fig. 6 and 7 is potentially interesting, but is not well integrated with the previous parts of the manuscript. Furthermore, it was unclear what point the authors are trying to make other than these neurons display sequential activity throughout all phases of the task. If the authors are trying to call attention to some special feature of neural activity in the maintenance phase, then it does not seem that the sequence analysis is the best choice. On the other hand, it was helpful to see that the miniscope data with single neuron resolution agreed with the photometry data (i.e. Fig. 6f agrees with Fig. 2b). But as noted above there is a question about the animals' behavior.

Minor:

1. The authors could have been more thorough in characterizing the neurons they are targeting. In particular, it would have been useful to know what percentage of neurons in mPFC they are labeling. Additionally, since the authors are recording and manipulating cell bodies rather than terminals, it seems important to know whether the mPFC to dmStr neurons project to other areas besides dmStr. They briefly touch on this issue in Lines 449-452, but it would have been nice to show the full projection profile of these neurons or discuss any literature may have already done this. Alternatively, do the authors have any evidence that it's specifically the projection to the striatum which mediates the WM maintenance effects reported here?

2. The behavioral performance in Fig. 5 is highly variable across the different groups, making it difficult to fully accept the effects of opto-stimulation. For example, in panel 5d, left the mean performance is 60% both off and on laser. But in panel 5d, middle the mean performance is ~45% off laser and increases to ~65% on laser. Why was the off laser performance different?

3. Page 3, Line 56: may be [better to say "interactions with MD nucleus in the thalamus..."]

Responses to Reviews, Nature Communications, NCOMMS-21-44417-T

Dear Reviewers, dear Editors,

We thank the reviewers for their careful reading, their specific comments on our work, and their constructive suggestions to improve our manuscript. We have added new experimental data, extended the data analysis substantially, and revised text and figure parts accordingly. We have also revised the abstract to convey the main message more clearly. Here, we first address the three major overlapping concerns that several reviewers raised. Below, we then reply point-by-point to each reviewer's comments and explain how we have addressed the specific concerns. In the revised manuscript, we have marked those sentences and sections where changes were made in yellow. We believe that we have addressed all major concerns and hope that you will find our revised manuscript much improved.

The **3 major concerns** that were raised by several reviewers are:

- 1. Behavioral variability and potential movement strategies for solving the task (R#2, R#3)**
- 2. Potential confounds of photometry signals due to behavioral variability (R#3)**
- 3. Variable baseline performance and variability across mice in optogenetic experiments (R#1-3)**

Reply to concern 1: Behavioral variability and potential movement strategies for solving the task (R#2, R#3). The reviewers raised the concern that it was not clear in the original manuscript whether different mice used similar behavioral strategies and whether some mice may have used some strategy to solve the task via stereotyped motor behavior. We addressed this concern by tracking mice with DeepLabCut (DLC) in all video recordings, particularly focusing on the maintenance period of the T-maze alternation task (see revised Methods). Briefly, from the coordinates of the mouse's nose, ears, and tail base (estimated in each video frame), we calculated geometrical and behavioral parameters potentially relevant for the T-maze alternation task: 1) the angle of mouse orientation (nose-to-tail base) relative to the axis of the T-maze main corridor (from start-box to T-junction); 2) the number of turns during the maintenance period, with a turn defined as changing head direction from facing the T-junction to facing opposite or vice versa (separately counting clockwise and counterclockwise turns). In addition, we quantified 3) the momentary position in the maze, 4) positional changes of the nose (frame-to-frame displacement divided by frame interval, thus comprising both running speed and speed of head movements); and 5) the time of the first re-orienting turn towards the door after arrival in the start-box. All these behavioral variables could potentially carry explanatory power of the future behavioral choice and/or the measured prefrontal calcium signals.

Our additional analysis did not reveal any indication of mice using stereotypic behaviors during the maintenance phase that would be predictive of their future choice. In the revised manuscript, we show these results as **additional panels c-h in Supplementary Figure 1**. In brief, we first compared the number of turns and mouse orientation (angle) during the maintenance period for correct Left vs. Right Sample Runs (Right vs. Left turn choices). To account for a possible dependence of correlations on trial time (*i.e.*, orienting towards the start box at the beginning of the maintenance period but orienting towards the T-junction at the end), we considered these 2 general orientations separately. Neither the trial-average of turn direction (after assigning +1 for each turn in a clockwise direction, -1 for each turn in counterclockwise direction) nor the trial-average of orientation angle during the maintenance period were significantly different for future Left vs. Right choice runs (in all n=6 mice). Specifically, we calculated average number of turns and average orientation per trial and pooled the results across all mice (bar plots in **Supplementary Figure 1c,d** and left panels of the polar histogram plots in **Supplementary Figure 1e,f**). In

addition, we show data for each mouse separately (orange circles connected by gray lines in **Supplementary Figure 1c,d**, and right panel of the **Supplementary Figure 1e,f**). When we tested each mouse individually, we found in one mouse (m6) a weak significance for both number of turns and orientation (after correction for multiple comparisons, actually the p-value was above 0.05).

New panels c-f in Supplementary Figure 1. Average direction of turns (c, d) and mouse orientation angle (e,f) during the maintenance period do not predict future choice. Results are shown separately when mice were oriented towards the T-junction and when they faced the opposite direction (towards the start box). For full legend see the revised manuscript.

In a second approach, we used all behavioral variables 1-5 to assess more complex behavioral strategies during the maintenance period that potentially could have been utilized to solve the task. For the dynamic variables 3 and 4 (position and positional changes), we resampled their temporal dynamics to a common duration (median duration of maintenance period across all trials; similar to the resampling of calcium signals). We reduced the dimensionality of the concatenated vector of these behavioral variables with Uniform Manifold Approximation and Projection (UMAP, **Supplementary Figure 1g**; see figure below) and tried to identify differences for Left vs. Right trials (**Supplementary Figure 1h**). We did not observe any behavioral difference by labeling Left and Right trials on the UMAP X-Y plot with blue and red, respectively. However, there was a gradual variability of behavior across mice: for example, m3 and m4 showed more similar behavior as compared to m5 and m6. When we inspected the UMAP X-Y embedding for every mouse separately we still did not find a separation of trials into Left vs Right (data not shown).

Taken together, we could not find any indication for mice utilizing specific movement strategies during the maintenance period to solve the task.

New panels g,h in Supplementary Figure 1. Embedding of all behavioral variables into a reduced dimensionality plane with UMAP could not explain left vs right future choices. For full legend see the revised manuscript.

Reply to concern 2. Potential confounds of photometry signals due to behavioral variability (R#3). To test for potential confounds in the photometric calcium signals related to animal movement, we correlated calcium signals during the maintenance period with behavior variables using linear regression. We particularly focused on positional changes (speed) as a parameter reflecting the overall amount of movement during the maintenance period. The overall correlation is reflected in the R^2 that determines the proportion of variance in the calcium signal that can be explained by the behavioral variables. Respective R^2 values were low with high p-values (*i.e.*, speed full trial $R^2= 0.001$, $p = 0.23$; speed towards Start Box $R^2= -0.002$, $p = 0.74$; speed towards T-junction $R^2= 0.001$, $p = 0.21$). Among all other behavioral variables only the orientation angle towards the T-junction was positively correlated to the calcium signal, but it potentially accounted for only 2% of variance ($R^2= 0.021$, $p = 0.015$ Bonferroni corrected).

To identify behavioral stereotypes or strategies, we used the same behavioral analysis with UMAP embedding as described above in **Supplementary Figure 1g,h**. This time we selected only 'Left Sample Correct' trials (*i.e.*, future right turns at the T-junction) and tried to cluster data with *k-means* in the UMAP X-Y plot. For *k-means* we tentatively asked to identify 3 clusters on the UMAP plane (**Supplementary Figure 2c**; see figure below). Silhouette plots reflect the cluster independence from its neighbors (**Supplementary Figure 2d**), for example cluster 2 has an overlap with both cluster 1 and 3. Therefore, we think that our behavioral data cannot be considered as separate clusters but rather represent a continuum of behaviors with gradual differences. However, for the sake of comparing calcium signals across potentially different behavioral groups, we relied on the labeling given by these three clusters. Indeed, behavioral variables differed across clusters, most prominently the position in the maze and the movement of the mouse (**Supplementary Figure 2e**). Next, we used trial labels for clusters 1-3 and asked whether calcium signals during maintenance were different across clusters. We found that calcium signals in cluster 1 were transiently elevated during the first half of the maintenance period (**Supplementary Figure 2f**; $p < 0.05$ for every time sample, Wilcoxon rank sum test). In addition, movement towards the T-junction was enhanced in this cluster as compared to other clusters. To make sure these two variables did not coincide in time, for every trial we calculated the proportion of time when mice started to move towards the T-junction (the end of the maintenance period). We plotted this variable as a histogram (bottom **Supplementary Figure 2f**). When mice were moving towards the T-junction (dashed line above ~80% of the trial duration) the calcium signals in all three clusters overlapped and their levels were significantly lower as compared to their peak values during the middle of the maintenance period. Also, the median of the z-scored calcium signals during the maintenance period did not differ across clusters

(Supplementary Figure 2j). In conclusion, we did not find obvious relationships between motor-related behavioral variables and the photometric calcium signals.

New panels c-j in Supplementary Figure 2. Elevated calcium signals during the maintenance period could not be explained by the motor-related behavioral variables. For full legend see the revised manuscript.

Reply to concern 3. Variable baseline performance and variability across mice in optogenetic experiments (R#1-3). We agree that mice showed variable task performance across sessions and that in our original manuscript baseline performance was variable. We have addressed these concerns by conducting additional experiments, which helped to better equalized baseline variability, and by additional data analysis. Specifically, we added experimental data from 3 additional mice to the datasets with pathway-specific ChR2 activation and MK801 pharmacological intervention together with ChR2 activation (MK801+ChR2). We also added 6 animals to the GFP control group. Furthermore, according to the suggestion of Reviewer #3, we separately analyzed Left and Right trials. See also our point-by-point responses, where we show the revised figure panels for ChR2 activation, MK801 plus ChR2 activation, ArchT inhibition, and GFP controls.

Regarding the day-to-day variability we like to point out that the optogenetic experiments were designed such that on each day only one of the periods of the T-maze task was targeted (*i.e.*, the laser was 'On' in 50% of randomly chosen trials either during encoding, maintenance or retrieval, as chosen by the experimenter at the beginning of each session). Therefore, variability of task performance across days presumably impacted the baseline performance during laser 'Off' sessions and the comparison with subsequent laser 'On' sessions. To quantitatively compare variabilities, we first used the Bartlett test to analyze whether the variance across all groups was different in experiments with ArchT inhibition. Variances were not different for laser Light 'On' vs 'Off' conditions in the distributions for all 3 groups (encoding, maintenance, and retrieval; p-values of 0.2, 0.29, and 0.19, respectively). Next, we tested whether the variance in encoding, maintenance or retrieval period was different for the Light 'On' vs 'Off'

condition, revealing a non-significant p-value of 0.09. We repeated the same analysis for the ChR2 activation experiment and for the MK801+ChR2 experiment. We found no significant difference in variances across all conditions except for ChR2 vs MK801+ChR2 ($p < 0.01$ for all periods, encoding, maintenance and retrieval). We think that the higher variability for the MK801+ChR2 group might be due to the MK801 pharmacological intervention, since even well-controlled doses of MK801 (ml/bw) could variably impact behavior.

In addition, we have tried to explain day-to-day variability of task performance with several hypotheses: 1. A learning or training dependence, *i.e.*, the performance on a specific experimental day would depend on the task performance on previous days; 2. A dependence on mouse motor behavior, *i.e.*, variables such as speed, number or direction of turns, or angle of head orientation could explain low vs. high task performance. To examine whether training history (expressed as changes of performance relative to previous training days) could explain the variance in task performance on a specific day, we pooled experimental datasets from ChR2, ArchT, and control groups, respectively. For every session we calculated the performance for Left Sample runs (for datasets with optogenetic perturbations we only used the Laser 'Off' group) and calculated the number of days passed relative to the last preceding training day. We found no correlation between these variables (explained variance $R^2 = 1.4\%$ and non-significant $p = 0.09$). Hence, we conclude that the training history did not contribute to the day-to-day variability in performance.

To examine whether motor behavior of mice could explain variations in performance, we used the behavioral variables estimated from DLC tracked video files. We tested whether the task performance could be explained either by the magnitude of averaged behavioral variables, or by variance across trials for every behavioral variable within session during correct Left Sample run trials, or by differences of behavioral variables between Left Correct and Left Mistake trials. Explained variance (as estimated by the R^2), was low ($< 4\%$) and non-significant for all behavioral variables either as regressed to the average, the variance, or the difference of behavior between correct and mistake trials (Extra Figure 1). We conclude that motor behavior parameters alone could not explain variability of the task performance.

Overall, we conclude that neither motor behavior nor long-term learning-related changes could account for the day-to-day variability of the task performance. We think that identifying the origin of the behavioral variability is challenging and would require further in-depth studies.

Point-by-point responses (reviewers' comments in *italics*, our replies in **bold**)

Responses to Reviewer #1:

Using fiber photometry and miniscope imaging, Chernysheva and colleagues reported three sets of experimental data in support of the main conclusion that striatum-projecting prefrontal cortex neurons support working memory maintenance. (1) there was strongest neural activity during the delay (maintenance) phase of WM in mPFC→dmStr projection neurons; (2) Optogenetic inhibition of mPFC→dmStr neurons selectively during the delay phase impaired performance while optogenetic activation of these labeled neurons during the delay alleviated WM impairment induced by MK-801; (3) miniscope imaging revealed the increased fraction of neurons (but not amplitude) during WM maintenance with apparently sequential activation patterns. The experiments were carefully designed and skillfully executed and the data analysis was sophisticated and apparently sound. These findings extend the previous studies and reinforce the conclusion, albeit not entirely novel, on the critical role of mPFC neurons in the working memory maintenance by isolating these mPFC→dmStr projection neurons and on sequential firing of mPFC→dmStr neurons to support WM maintenance.

We thank the reviewer for the appreciation of our work. In the context of previous studies on this topic, we indeed see the focus on specifically characterizing the mPFC→dmStr projection as the novel and complementary aspect of our work.

However, there are several major concerns that need to be addressed before the manuscript to be considered for publication in NC.

1. *Because ChR2 activation during the maintenance period produced only a small increase (about 3%) in WM performance, the authors suspected that the high (~80%) baseline performance may have masked ChR2-mediated potential improvement. To address this issue, they used MK801 to impair WM (below 70%) and then showed that ChR2 activation during the delay (but not the sample and choice) phase increased WM performance. However, the baseline performance (“Off”) in these three phases were quite different (~60% sample, 45% delay and 70% in choice). The preferential improvement by ChR2 in the delay phase may be confounded by its particularly “low” baseline. The authors need to carefully address this issue to support the conclusion.*

Reply: We addressed this question by adding 3 additional mice with ChR2 activation to the dataset (now n=9) and by performing the additional analysis described in our reply to General Concern 3. The baseline is now more homogenous across conditions of encoding, maintenance, and retrieval periods, with overlapping standard errors of the mean and indistinguishable variance across all conditions as assessed with Bartlett's test. We also treated Left Sample and Right Sample Runs separately.

We have updated Fig. 5c,d accordingly, now showing the performance results for Left Sample Runs in this main figure (left plot in figure below). For your information, we also show here the corresponding plot for Right Sample Runs (right plot below), which did not show any significant differences as we mention in the figure legend.

Left: Revised panels c,d in Figure 5. WM effects of optogenetic activation of the mPFC→dmStr pathway. (c) Optogenetic activation of the mPFC→dmStr pathway during the encoding (E), maintenance (M), and retrieval (R) periods in Cre-dependent Chr2-expressing mice (n = 9 mice) did not change performance during any period of the task (Bonferroni corrected p-values for Left Sample Run trials: E: 0.9 M: 1.71, R: 2.28; Right Sample Run trials performance also did not change in all periods of the trial: Bonferroni corrected p-values E: 2.12 M: 0.16, R: 0.38; Wilcoxon signed rank test). (d) Optogenetic activation of the mPFC→dmStr pathway partially rescued WM impairment induced by MK-801 injection when applied during the WM maintenance period (performance increased from $48.1 \pm 7.8\%$ to $65.9 \pm 7.2\%$; Left Sample Run trials, Wilcoxon signed rank test, Bonferroni corrected M: * $p = 0.011$, E: 1.38, R: 2.22; Right Sample Run trials performance also did not change in all trial periods; Bonferroni corrected p-values E: 1.96 M: 1.71, R: 1.03). For a given task period, Chr2 activation was performed in 50% of trials in a random fashion. Black lines represent individual sessions. Error bars show s.e.m. Grey shaded error areas represent the s.e.m. spread during Light Off Encoding.

Right: Corresponding plots for the Right Sample Run trials. None of the comparisons with Laser Light 'Off' vs 'On' revealed a significant difference. *We have not included these plots in the revised manuscript but could do so if wished for.*

2. *The strongest increase in calcium signaling is at the event 5, the turning of T maze. This calcium signal increase can have alternative interpretation, i.e. calcium signal response to position cue (i.e. the turning point). Such stimulus-trigger increase in calcium signal in response to position cue has been shown in the recent recording of calcium in astrocytes and neurons (see Lin et al. Entrainment of Astrocytic and Neuronal Ca²⁺ Population Dynamics During Information Processing of Working Memory in Mice. Neurosci Bull. 2021 Oct 26). This needs to be carefully addressed.*

Reply: We thank the reviewer for pointing out this interesting recent study. We carefully looked into this issue by plotting the calcium signals surrounding event 5 (Extra Figure 2 below; ± 1 s, left panel; >5 s, right panel). Event 5 was defined as the time point when the mouse passed the ROI in the main corridor close to the T-junction, i.e., after the mouse fully completed the turn back towards the startbox. Because at this time animals need to start maintaining information about their run history, we also used event 5 to define the start of the maintenance period. We did not observe any obvious neuronal calcium signal in mPFC related to the turn before event 5. Only during the first second after event 5, the calcium signal started to gradually increase, without yet being discriminative for correct vs. mistake trials. In the

following seconds the calcium signal then further increased and clearly diverged for correct vs. mistake trials (Extra Figure 2, right).

We cannot compare these results directly to the results of Lin et al. because we have not measured astrocytic calcium signals (which is the main focus of the Lin et al. study) and Lin et al. do not show neuronal calcium signals for mPFC (but for hippocampus). Nonetheless, we now cite this interesting paper in the discussion (pg. 21, line 517), emphasizing the need to also consider and study other possible circuit mechanisms of WM.

Extra Figure 2. Photometric calcium signal around event 5, which occurs after the turn when the mouse entered the main corridor (what we define as the start of the maintenance period). Left: mean calcium signal aligned to event 5 for all Left Sample run trials (red: correct trials, $n = 704$; black: mistake trials, $n = 346$; 6 mice and all expert sessions). Event 5 is defined in the ROI indicated when the animal has fully turned its body into the main maze arm. Right: mean calcium signal after event 5, shown up to 6.5 s into the Maintenance period, for all Left Sample run trials (blue: correct trials; black: mistake trials; as in the left panel). *We have not included this Extra Figure 2 in the revised manuscript but could do so if wished for.*

Furthermore, the standard definition of the delay phase starts at the close of the rest chamber (event 6). There was no further calcium increase in event 6 (compared to event 5). Therefore, the main increase in calcium signaling seem to occur before the maintenance. This requires careful evaluation and interpretation. It is important to dissect out the contribution of “position cue” to the calcium increase in WM information processing.

Reply: As clarified above, the ‘position cue’ of the turning point occurs in our setup just before event 5 and we did not observe any obvious calcium signal in mPFC→dmStr neurons related to this cue. In our view, event 5 (when the animal is fully back in the main maze corridor) represents the start of the maintenance period. At this time, the animal has no access to the T-maze arms anymore (doors are automatically closed) and it therefore has to start maintaining information about the previously visited arm in WM. To us, it therefore makes sense that the mPFC starts rising at this time point and not only once the animal is back in the start box.

Surveying previous studies using a similar task design, we found that the experimental procedures vary and that, unfortunately, the start time of the maintenance period often is not precisely defined. In one

variation of the task, the animal is actually taken out of the sample arm and brought back to the start box in a transfer cage. In this case, the delay period starts when the animal is taken out of the sample arm, which is similar to our definition. The Lin et al. paper also appears to use a similar definition as us (“when the mice returned to the start arm at the beginning of the maintenance phase”, their Methods section). In yet other studies, infrared sensors were placed at a similar position as our ROI “event 5”, thus again detecting when the animal has fully returned into the main maze arm and thereby marking the start of the maintenance period (e.g., Liu et al. 2018, ref. 45). Overall, we agree that there is some confusion in the field and that we therefore must explain our definition and reasoning clearly. In the revised manuscript, we have therefore clarified our description of the event structure in the T-maze and how it is linked to the definition of the maintenance period (pg. 6, line 143-147).

3. It is important to provide the calcium signal data after MK801 and HCN-channel blocker J&J12e, so that possible correlation of these calcium signal can be made with their effects on WM performance.

Reply: We agree that such additional neuronal data during the pharmacological interventions would be interesting. Unfortunately, we have not collected such data and we are currently not able to conduct such additional experiments because our collaboration with Roche has officially ended so that the HCN-channel blocker J&J12e is not available for us anymore. In addition, we currently do not have an approved animal license to perform such experiments combining both compounds. Obtaining such approval has become more difficult in Switzerland and would take a substantial amount of time. Thus, we have no opportunity to easily collect such additional data.

4. While fiber optometry recording of calcium signaling neural populations show clear increase in the maintenance phase (5-7), miniscope of individual neurons show only a minor increase in the fraction of neurons that fired above the threshold during this phase. However their mean spike activity was in fact lower than that of other two phases. This needs to be acknowledged and carefully interpreted.

Reply: We think that the miniscope data matches the photometry data well. The miniscope data show that the fraction of neurons that are preferentially active during the maintenance period (0.56, i.e., 56%; Fig. 6f) is significantly increased compared to encoding and retrieval (both 0.22; Fig. 6f). At the same time, the mean spike rate is lower by 32% in the maintenance compared to encoding and retrieval period (spike rates in a.u. of 1.23 vs. 1.8; Fig. 6g). This suggests that the resulting signal during the maintenance period should still be ~1.74 times higher as compared to the encoding and retrieval period, in line with the photometry data. We like to emphasize that the estimation of mean spike rate based on a deconvolution algorithm is only approximate, especially when the activity patterns include large-amplitude calcium signals.

For clarification, we have rephrased the respective text passage (pg. 14, line 332-334) and added the exact numbers in the legend of Fig. 6.

Minor points (Reviewer #1):

The study by Liu and colleagues “Medial prefrontal activity during delay period contributes to learning of a working memory task, Science. 2014 Oct 24;346(6208):458-63” should be cited in the Introduction.

Reply: In the introduction, we now cite this study, which investigated WM using an olfactory delayed non-match-to-sample task.

Please provide more detailed descriptions of coupled analysis of event-specific calcium signaling with behavioral parameters (events 1-10) including apparatus, label, data analysis.

Reply: We have re-organized and augmented the Methods section by adding further details about the event labels (detection of specific positions in the T-maze), the synchronization of recordings with the control of the behavioral apparatus, and the analysis of calcium signals and behavioral variables (pg. 27-30).

Responses to Reviewer #2:

Chernysheva et al. studied striatum-projecting prefrontal neurons in mice during a spatial-alternation task. The data are well presented and the manuscript well-written. The experiments will be of interest to those studying mPFC->dmStr during behavior. Limitations in the behavioral task design and missing control experiments weaken some of the conclusions. Overall, it's unclear how the present results substantially advance understanding of working memory and the mPFC->dmStr pathway.

We thank the reviewer for their comments. In the revised manuscript we provide an extended behavioral analysis and hope that we could address most of the concerns satisfactorily.

1. 1.1 *The behavioral task design does not allow for strong conclusions. First, it's not clear different mice used similar behavioral strategies. Training occurred over a few days, and the variance in behavior is very large (Fig. 3d, Fig. S1a, Fig. 4b,c). Behavioral effects of mPFC silencing (9% performance decrease in Fig. 3d) are much smaller than differences across mice. At the very least, there should be a more thorough description of the behavioral data (e.g., showing learning curves, correlating them to effects of mPFC silencing).*

Reply: Regarding the first comment that “it is not clear whether mice used similar behavioral strategies”, we refer to our above reply to the General Major Concern 1. In brief, we tracked the mouse on videos with DeepLabCut and quantified several behavioral parameters (mouse orientation, the number and direction of turns, position in the maze, positional changes/movement, time of first re-orienting turn). We then applied dimensionality reduction of the behavioral data via UMAP embedding and tried to identify specific clusters corresponding to distinct behavioral strategies. However, we rather found a spectrum of behaviors with gradual differences across mice but with all mice contributing to a large range of the spectrum.

Regarding the second issue of variability, indeed the behavioral variance is large. We discuss this concern in our reply to General Major Concern 3 (see above). In brief, we attempted to explain large day-to-day variance in behavioral performance by training or learning effects (relative to the preceding sessions) or by motor-related variables (potentially reflecting different behavioral strategies). However, we rejected both hypotheses as we did not find significant contributions to explained variance in our dataset.

1.2 *Second, referring to the task as requiring working memory (WM) is an assumption not supported by the data. Mice could conceivably solve the task without WM (e.g., prepare a policy of left-right-left-right... in advance). I suggest revising the language to reflect this (e.g., "delayed alternation" instead of WM), although I appreciate that this is a term of art in the literature for this behavioral task.*

Reply: There might be a misunderstanding. Our task is not a ‘spontaneous alternation’ task but is rather best described as T-maze delayed non-match-to-place (DNMTP) task, where the direction of the sample run is pseudo-randomized and mice therefore cannot simply use an alteration policy. Mice had to experience the left or right sample arm during the encoding period to prepare their future policy uniquely for every trial. The randomization of sample runs was mentioned only in the Methods section of the original manuscript, and we apologize if the terminology has been confusing. For the revised manuscript, we decided to use the more appropriate term of “T-maze DNMTP task”, as it also has been used for example by Spellman et al. (ref. 15) or Lin et al. (new ref. 66). In addition, we now emphasize this aspect of the task design also in the main text (p.4, line 100-105).

2. *The relationships shown in Fig. 2d have very small R^2 values and thus should be interpreted with much more caution. It looks like the three rightmost points in each scatter plot contribute a lot of leverage to the regression. Does it survive removal of those points? Similarly, the interpretation that RT decreases with change in late-delay-phase activity implies higher decision confidence is dubious. RT and confidence are often correlated, but needn't be. This latter statement could be better supported with references.*

Reply: The correlation between task performance and z-score differences in the late maintenance period was still positive and significant after the removal of the three rightmost points. However, we fully agree with the reviewer about the interpretational challenge, and we therefore decided to remove this figure panel and the associated text from the manuscript.

3. *There are several relevant papers that would be useful to discuss in the context of the present results, including prior demonstrations of sequential activation in similar tasks in rodents (mostly rats).*

a. Brito et al., *Exp Brain Res*, 46, 52, 1982

b. Jung et al., *Cereb Cortex*, 8, 437, 1998

c. Baeg et al., *Neuron*, 40, 177, 2003

d. Fujisawa et al., *Nat Neurosci*, 11, 823, 2008

e. Horst & Laubach, *Neuroscience*, 164, 444, 2009

f. Horst & Laubach, *J Neurophysiol*, 108, 3276, 2009

Reply: We thank the reviewer for pointing out these references, some of them being classics for engagement of mPFC neurons during WM. We now cite these references in the introduction (pg. 2; line 48). In addition, we refer to several of them specifically in the course of the discussion (pg. 18, line 453; pg. 19, line 456).

4. *Were there spatial relationships between calcium dynamics of single neurons within a field of view? Others have observed layer-specific activity in this region. The imaging geometry here may allow for a laminar analysis.*

Reply: We agree that the spatial profile of activity patterns is an interesting aspect. We looked at this issue by analyzing the spatial distribution of neurons in the miniscope field-of-view, grouped according to their functional specialization. To this end, we used the same functional classification of neurons as in Figure 6, distinguishing maintenance cells from non-maintenance cells (grouping together the encoding and retrieval cells). These two functional groups were spatially intermingled and we did not find any obvious difference in their spatial distribution as illustrated for two example mice (m060 and m061; see Extra Figure 3 below; results were similar for the other mice, data not shown).

Unfortunately, it is also not possible from our data to clearly identify laminar borders. Overall, we think that based on our miniscope dataset we cannot draw firm conclusions about a potential spatial organization of distinct functional subgroups. We believe that revealing laminar differences and potential differences in spatial organization will require further experiments that specifically address these questions.

Extra Figure 3. Spatial distribution of neurons in the miniscope field of view for two example mice. We did not reveal any obvious spatial organization of neurons mainly active during the maintenance period compared to neurons mainly active in the other periods. We have not included this Extra Figure 3 in the revised manuscript but could do so if wished for.

5. *The distributions of durations of encoding, maintenance, and retrieval for correct choices and mistakes are interesting. The means are stated to be no different from each other, but it looks like the variance of the "encoding" phase is larger on mistake trials versus correct trials. Is that true? It also looks like there is a trend for the mistake trials to have shorter durations in the "retrieval" phase. Do these observations impact the interpretation of the neuronal data?*

Reply: Indeed, the variance of the durations in the encoding period was larger during mistake trials as compared to correct trials. This may indicate the different motivation levels of the mouse. In our study we focused on the importance of the mPFC→dmStr pathway for WM maintenance and did not analyze the photometric signals during the encoding period in detail. As the variance of the durations of the maintenance period did not differ between correct and mistake trials, we do not see any potential impact on our interpretation of photometric calcium signals during the maintenance period.

The reviewer is also correct that there is a trend for the mistake trials to have shorter durations of the retrieval period. We believe this effect presumably arises because mistake trials were not rewarded in our DNMTTP task. The shorter duration thus may simply reflect the absence of the reward and consequently also of the reward collection period.

6. *The conclusion that "...performance decrease was not simply induced by altered locomotion" (line 203) does not completely follow from the observation that mean duration of choice runs were similar (Fig. 3e). For example, different movement kinematics produced by bouts of increases and decreases in velocity would produce the same choice run duration as constant velocity. Particularly in a task in which movement preparation can be used to perform (e.g., without WM), this claim and the manuscript overall would benefit from a more thorough analysis of movements. The authors allude to this somewhat in the statement on lines 175-177, when analyzing ipsi- versus contralateral movements (Fig. S2).*

Reply: We agree that the previous conclusion was not justified based on the comparison of mean durations. Although in a home-cage test we did not detect differences of mouse movement and turn direction upon optogenetic activation (see revised Supplementary Figure 7a,b below; formerly Suppl. Fig. 6), we agree that a more quantitative analysis of animal behavior and movement patterns for the optogenetic experiments is necessary. To do so, we applied the same UMAP embedding approach as described in our reply to General Major Concern 1. We found minor changes of motor behavior induced by optogenetic light stimulation during the maintenance period of the DNMT T-maze task. Specifically, on the UMAP plots, 3 behavioral clusters were apparent (revised Supplementary Figure 7 below): Cluster 1 and 3 contained both Light Off and On trials ('mixed clusters') whereas cluster 2 was composed of Light On trials only. We tested all behavioral variables. The overall movement patterns and the trial-average direction of turns were not affected (panels d,e). The total number of turns was the only behavioral variable altered by light stimulation if pooled according to the cluster label (in the subset of Light On trials in Cluster 2, panel e; *** $p < 0.001$, Wilcoxon rank sum test). However, the change in the average number of turns was not significant if pooled by trial type, i.e., Light Off vs Light On in all Left Sample Run correct trials (panel d). Importantly this minor change of motor behavior did not affect the task performance (main Figure 5c), therefore rendering alteration of this motor variable not predictive of the upcoming choice. We added these findings as new panels in the revised Supplementary Figure 7.

Revised Supplementary Figure 7 with new panels c-e. Alterations of mouse behavior in the home cage (a,b) and during the maintenance period of DNMT T-maze task upon optogenetic activation of mPFC→dmStr pathway (c-e). For full legend see the revised manuscript.

We used the same approach also to compare Left Sample Run correct trials during the maintenance period with and without optogenetic inhibition. Similar to the results of optogenetic activation, the delivery of light for optogenetic inhibition during the maintenance period did not affect movement, trial-averaged direction of turns, and other motor behavioral variables, except for the average number of turns (tested in the Cluster 2 containing a subset of Light On trials, Extra Figure 4 below).

In summary, we found no obvious change in the overall motor behavioral motifs induced by optogenetic manipulations except for a slight decrease in the average number of turns (presumably reflecting shorter durations of the maintenance period). In addition, this change was apparent in only a subset of Light On trials.

Extra Figure 4. Optogenetic inhibition of the mPFC-dmStr pathway affected behavior. (a) UMAP embedding of behavioral variables during Maintenance period in ArchT expressing mice during Light On and Off trials. (b) Mouse movement, average number of turns and trial-average direction of turns during Maintenance period pooled by the trial type. (c) Same as b, but pooled by the tentative cluster assignment. We have not included this Extra figure 4 in the revised manuscript but could do so if wished for.

7. The effects of HCN blockade appear confined to the Maintenance period in Fig. 4a, but look more nonspecific in Fig. S4a (i.e., a DC shift for drug relative to vehicle). Please clarify this, and the degree to which these signals can be compared appropriately across sessions.

We agree with the reviewer that the effects of the compound (HCN blockade) appear not specific to the maintenance period. We mention this point now more explicitly in the revised main text (pg. 10, line 247). Fluorescence signals of calcium indicators must be normalized ($\Delta F/F$ or z-scored) to a defined baseline because absolute fluorescence values cannot be compared meaningfully within and across dataset. As our main interest was to compare the changes in fluorescence signals (as proxy of underlying spike rate changes) in the maintenance period, we chose the signal in the 1-s time window before event 5 (start of maintenance period) as the common baseline for normalization (z-scoring). The reviewer correctly points out that relative to this 'pre-maintenance' baseline, fluorescence signals in encoding and retrieval period also appear to be enhanced, especially for the reward periods 3-4 and 9-10.

8. The effects in Fig. 4c are interesting and appear bimodal. That is, it appears that mice that started off worse at the task had little-to-no effect of MK-801+J&J12e, whereas those that had higher initial performance had a substantial drop in performance with the combination of drugs (apparently not as bad as with MK-801 alone). First, is this true? If so, does it affect the interpretation (e.g., there could be two strategies of solving the task, one involving WM and one not)? Second, please clarify whether the statistical tests in Fig. 4c included omnibus test (e.g., ANOVA followed by planned contrasts).

Reply: Indeed, 4 out of 11 mice in Fig. 4c minimally reduced the performance during MK801+J&J12e administration. To assess whether the distribution was bimodal we calculated Warren Sarle's bimodality coefficient. If bimodality coefficient is greater than 5/9 (~0.555 – the value for the uniform distribution), the underlying distribution might be bimodal or multimodal. The bimodality coefficient for the difference of task performances between group 3 and 4 in Fig. 4c is 0.47. This value is lower than the criterion for bimodality, rejecting the hypothesis, which however could be due to the relatively low number of samples. For the statistical tests, we used Bonferroni correction which we now indicate in the Fig. 4c legend.

9. The ArchT inactivation experiment (Fig. 3) requires a fluorophore-alone control.

Reply: We performed additional control experiments in n= 6 mice expressing GFP fluorophore in the mPFC→dmStr pathway. We found no significant difference in task performance for Light On vs Off conditions (new panel in Supplementary Fig. 4). As in the ArchT experiments shown in main Figure 3, light stimulation was applied in random 50% of trials, targeting only one specific task period in each session.

Left: New panel d in Supplementary Figure 4. GFP control experiments to test for potential light-induced behavioral effects in ArchT experiments. No significant effect on performance was observed upon light stimulation in control mice expressing GFP instead of ArchT in the mPFC→dmStr pathway (n = 6). Data are from Left Sample Run trials. Right Sample Runs also revealed no significant effect (data not shown). Black lines represent individual mice. Dashed lines, 50% chance level. Standard error of mean (s.e.m.) overlapped across all conditions, as exemplified by gray bars representing s.e.m. for the maintenance period.

Right: Corresponding plots for Right Sample Runs in the GFP control experiments (n =6). None of the comparisons with Laser Light 'Off' vs 'On' revealed a significant difference. We have not included this plot in the revised manuscript but could do so if wished for.

10. I'm unclear about whether the BDO shuffle control is appropriate. Depending on the sampling distribution of , using a permutation test to compare to shuffled data could come with dangerous assumptions (why not bootstrap?). In any case, wouldn't simply cross-correlating $x(t)$ for neurons i and j , with the data split in half for cross-validation, be appropriate here?

Reply: We initially attempted to infer the order of neuronal activity by calculating trial-wise cross-correlation followed by lag extraction. However, there were some challenges associated with this approach. Firstly, single-trial cross-correlation functions often had multiple peaks (e.g., related to multiple calcium transients during a trial). The inference of a single lag from such cross-correlation traces thus would not be robust. Secondly, our experimental design with freely moving animals results in trials of variable duration. Because of this, there is no clear way to accumulate the inferred lags across trials, as their absolute and relative times were frequently different in different trials. Finally, we were concerned that cross-correlations could over-emphasize noise correlations over large but temporally confined spike-related activity.

We designed BDO as a principled approach to overcome the above challenges. As an integral metric, it is far less sensitive to the exact position of peaks and noise. Further, since a single binary variable is computed for each trial, confidence intervals of its trial-average can be readily estimated without concerns of autocorrelation effects. We have extensively tested the behavior of BDO on simulated data and confirmed that it can correctly discriminate between orderable and random data, as well as correctly rank data with different underlying degrees of order.

The reviewer also suggested that we use bootstrap instead of permutation testing. We agree that bootstrap covers a larger random domain by varying the quantity of each datapoint in addition to their order. However, we note that bootstrap is also vulnerable to biases.

Minor comments (Reviewer #2):

1. *The sentence on lines 50-52 could be qualified by a statement such as "...major axonal projections include..." or a reference to species differences. For example, mPFC across mammals projects strongly to other neocortical areas, hippocampus, and other midbrain structures than VTA.*

Reply: We agree and edited the text as suggested.

2. *I'm not sure I see how the experiments test "the computational role of the mPFC->dmStr activity" (lines 72-73).*

Reply: We edited the text. The sentence now reads: *"Using optogenetic and pharmacological manipulations, we tested the functional significance of the mPFC → dmStr pathway activity during specific periods of the WM task."* (pg. 3, line 78)

3. *What produced the decreased GFP signal in Fig. 2a? Does this impact the interpretation of the decreased GCaMP signal during those two time periods?*

Reply: The transient fluorescence decreases were rather small and could not explain the difference observed in Figure 2c. What produced these decreases we are not sure about. It could be related to motion artifacts or hemodynamic signals during the reward consumption (phases 3-4 and 9-10), as we now mention in main text (pg. 7, line 164). Because the mean values per period were rather uniform for the GFP signal (bottom plot Fig. 2b), we do not think they would affect the profile of mean GCaMP6m signals across periods.

In the Methods, we now included a whole paragraph on the control experiments for fiber photometry signals (pg. 30, line 745-757). We gain further confidence about the neuronal origin of photometric signals from the miniscope experiments. There, the cellular resolution allowed us to pool fluorescence signals only from identified cell bodies, reducing potential confounds of hemodynamic signals. The

resulting calcium signal averaged across all cells in the field of view was very similar to our photometry bulk recording (Supplementary Figure 8).

4. Please show the data underlying the means and SEMs in Fig. 3c (e.g., scatter plots, or line plots underneath, as in Fig. 3e).

Reply: We included additional panels on Figure 3c and 5b showing the distribution for change of firing rate relative to the baseline (laser Off) for all sites on the electrophysiology probe. Spiking rates of neurons at different sites were variably affected by the optogenetic perturbations.

5. The "Photometry analysis" section in the Methods would benefit from a more thorough description of control analysis (i.e., GFP channel).

Reply: We included a more detailed description of the control analysis in the Methods section, describing the dual-wavelength excitation approach and the GFP control experiments (pg. 30, line 745-757).

6. The very large diameter implants (0.4 mm or 1.2 mm cylinders) clearly caused substantial damage to the areas dorsal to PrL. Does this affect the interpretation of the data?

Reply: Indeed, both fiber and GRIN lens implantation cause damage to the regions dorsal to PrL, including a small part of secondary motor cortex and areas of cingulate cortex. However, because of the more horizontal orientation of PrL neurons we assume that the local circuitry is rather intact. At least the implants of different diameter (photometry fiber 0.4 mm vs miniscope GRIN lens 1.2 mm), presumably causing variable degree of damage to the tissue, still showed consistent task-related activity patterns. We acknowledge that we cannot entirely exclude some effects of the implants on larger-scale circuitry.

Responses to Reviewer #3:

The authors have studied the role of striatal projecting prefrontal neurons (mPFC to dmStr) in guiding working memory behavior. Their study employs pathway-specific fiber photometry, optogenetics, pharmacology, and miniscope calcium imaging in mice performing a T maze alteration task. They find that neural activity is greatest during the maintenance period of the task, in which mice must maintain a working memory of the correct choice. They also find that perturbing neural activity selectively during this maintenance period produces the greatest change in behavior. From these results the authors conclude that mPFC to dmStr projecting neurons play an important role in maintenance of working memory. The results are noteworthy for shedding light on the role of corticostriatal neurons in regulating working memory. Furthermore, the result that these neurons appear to play the most important role during the maintenance period is novel and likely to interest the field. However, the lack of detailed behavioral characterization in the WM task made it difficult to fully accept the results as presented without ruling out other possible interpretations. The manuscript is clearly written for the most part, and this work presents data which have the potential to significantly interest the field, but the reviewer felt that some crucial details to rule out potential flaws were missing. Furthermore, it wasn't clear how the sequential activity analysis fit in with the rest of the study.

We thank the reviewer for their assessment of our results as noteworthy and novel. In the revised manuscript we provide a more comprehensive characterization of behavior, and we have clarified how we see the analysis of sequential activity fitting in this study.

Major comments:

1. 1.1 *The photometry results in Fig. 2 are interesting, but it's important to get some context about what the mice are actually doing during the maintenance phase of the task. Specifically, it seems important to know whether, during the delay period, mice sit still, or tend to rotate in the direction of their intended choice. For example, if they have to make a left choice, do they tend to show more leftward rotations during the delay period on correct versus incorrect choice trials? If so, this would present a potential confound in the photometry and miniscope results.*

Reply: We refer to our reply to the Major General Concern 1, where we quantified mouse behavior during the maintenance period and could not identify behavioral parameters that would predict correct future choices to turn Left or Right at the T-junction. We hope this additional analysis already addressed the reviewer's questions about the mouse behavior during the maintenance period.

Triggered by the reviewer's comment, we here further extended our analysis to compare correct and mistake (incorrect) trials (Extra Figure 5 below). For all 405 Left Correct, 136 Left Mistake, 407 Right Correct, and 132 Right Mistake trials we calculated the average number of turns and orientation angle. As in our reply to Major Concern 1, we considered the two general orientations separately (towards the T-junction, top row; towards the start box; bottom row). We show results for Left Sample runs only. Pooled across all mice (n=6) and all trials, neither the average turn direction nor the average orientation angle differed significantly for Left Correct vs. Incorrect trials (barplots on the left; mean \pm s.e.m.). In addition, we show data for each mouse separately (orange circles connected by gray lines, left plot; right plots for orientation angle). Only one mouse (m2) showed a weak tendency for a difference in the average orientation angle in Left Correct vs. Incorrect trials (mean 3.7° in Correct, mean 1.1° Mistake; orientation towards the start box). However, once corrected for multiple comparisons, the p-value was >0.05 for this mouse, too. Results for Right Sample runs were similar, with no difference in terms of the average number of turns and the orientation angle (data not shown). Thus, as for Correct Left vs. Right choices, we also could not identify specific behavioral motifs that would be predictive of Correct vs. Mistake choice runs.

Extra Figure 5. Average direction of turns (left panels) and average mouse orientation (middle and right panels) for correct and incorrect Left Sample Runs. Neither of these two behavioral parameters, assessed during the maintenance period, did predict future choice. Results are shown separately for when mice were oriented towards the T-junction (top row) and when they faced the opposite direction (bottom row). We have not included this Extra Figure 5 in the revised manuscript but could do so if wished for.

1.2 Similarly, the optogenetic inhibition data in Fig. 3 are interesting, but it's important to know whether the laser has any effect on the animal's motion during the maintenance phase itself. For example, if mice tend to rotate in the direction of their intended choice, but the laser impairs this rotation, that presents a potential confound. This concern would largely be addressed if the mice just sit still or are generally not very active during the maintenance period.

Reply: According to the reviewer's suggestion on the Left Sample Run trials mice would have increased clockwise turns to exercise the future choice turn to the right and the ArchT mediated inhibition of the mPFC→dmStr pathway would suppress such turns. We performed a similar behavioral analysis as in reply to point 1.1, but now for two groups representing trials with the inhibition Laser On and Off. We separately analyzed correct and mistake (incorrect) trials, and we found no such bias for both turn direction and orientation angle (Extra Figure 6 below). In this dataset, however, mice showed a general bias towards counterclockwise direction (~1 turn out of ~10 balanced turns), irrespective of the trial type. Also, we noticed that the variance of average orientation angle in mistake trials was larger during trials with inhibition Laser On as compared to mistake trials with Laser Off (Extra Figure 6b, Left Mistake).

To identify other possible changes of behavior during inhibition we used all behavioral variables and performed dimensionality reduction with UMAP (Extra Figure 4 above; reply to point 6 of reviewer #2). We found that Left Sample Run correct trials were partially segregated in Light On and Off trials. To

identify variables contributing to this partial segregation, we tested all motor behavioral variables separately (for Laser On vs. Off group, panel d; Cluster 1 vs. 2 group, panel e). Overall we did not find differences in movement, trial-averaged direction of turns and other variables, except for the average number of turns (when tested in the cluster 1 vs 2, $*p < 0.5$ Wilcoxon rank sum test).

In summary, we see no change in the overall motor behavior except for a slight decrease in the average number of turns during Light On trials. This change was apparent in only a subset of Laser On trials and did not reach significance for Light On vs Off group (Extra Figure 4b).

Extra Figure 6. Average number of turns for 'Left sample run' trials compared for Laser Off and On trials. We have not included this Extra Figure 6 in the revised manuscript but could do so if wished for.

- Fig. 3d is a very important and potential interesting result, but several concerns were noted. First, the authors did not provide exact p values for all plots. Second, on line 200 they state $p = 0.0035$ yet in the figure legend they show one star for significance ($* p < 0.05$). This needs to be clarified. Third, the source of high variability of the data in the middle panel (light on during maintenance) compared to the left and right panels was unclear and a bit concerning. This led the reviewer to wonder whether the authors are not capturing some important feature of animal behavior that is taking place during the maintenance phase. Finally, it appears the authors are pooling across the ipsilateral and contralateral turn trials, but could they separately check whether performance is differentially affected on ipsi vs contra trials?

Reply: We agree that the description of these results was insufficient. The reviewer is correct that we had pooled ipsi- and contralateral choice runs in the original manuscript. In the revised manuscript, we have now analyzed Left and Right Sample Runs (corresponding to Right and Left Choice Runs) separately. Consistent with the photometry results (main Figure 2c,d), photoinhibition had a differential effect on ipsi- vs. contralateral Choice Runs: Silencing mPFC neurons during the maintenance period reduced the task performance only during ‘Left Sample Run’ trials. In the revised panel d of Fig. 3 (see below) we therefore now show the results for Left Sample Runs. For comparison we also show the results for Right Sample Runs below. In the legend to Fig. 3d, we now also provide the exact p-values for all comparisons (including the Right Sample Runs).

In response to the third point of the reviewer (high variability in middle panel) we tested for differences in the variance for all periods using Bartlett’s test. We did not find any significant differences for encoding, maintenance and retrieval periods ($p > 0.1$). To visualize the overlap of variances, we plotted the s.e.m. range for the encoding Laser Off period as horizontal shaded error bar. This bar overlapped with the s.e.m. ranges of all groups except for the maintenance inhibition in Left Sample Runs. The larger s.e.m. in the joint ipis/contra plot in the original manuscript was due to the variable performance during ‘Right Sample’ trials in two mice.

We hope that the description of the photoinhibition results is now more convincing.

Left: New panel d in main Figure 3. Photoinhibition of mPFC→dmStr activity during WM maintenance impairs task performance. Left Sample Run trials are shown on the inset d and emphasized with the black frame. For comparison, we also show Right Sample Run trials (right inset, not included into the main figure). Optogenetic inhibition of mPFC→dmStr decreased the task performance specifically during maintenance period and during trials requiring contralateral turn.

Right: Corresponding plots for Right Sample Runs with photoinhibition ($n = 6$). None of the comparisons with Laser Light Off vs On revealed a significant difference. *We have not included these plots in the revised manuscript but could do so if wished for.*

3. *The sequential activity analysis in Fig. 6 and 7 is potentially interesting, but is not well integrated with the previous parts of the manuscript. Furthermore, it was unclear what point the authors are trying to make other than these neurons display sequential activity throughout all phases of the task. If the authors are trying to call attention to some special feature of neural activity in the maintenance phase, then it does not seem that the sequence analysis is the best choice. On the other hand, it was helpful to see that the miniscope data with single neuron resolution agreed with the photometry data (i.e. Fig. 6f agrees with Fig. 2b). But as noted above there is a question about the animals’ behavior.*

Reply: Because neuronal sequences are a fundamental concept of neural dynamics, especially regarding short-term memory, our goal was to reveal whether the subpopulation of mPFC→dmStr projection neurons shows temporally structured firing patterns, i.e., sequential activity. We think that many readers also will wonder about the neuronal population dynamics underlying the bulk activity signals measured by photometry. With the cellular resolution of miniscope imaging we could start dissecting the different aspects of population dynamics (fraction of neurons active and temporal profile of single-cell activity). We further like to emphasize that besides finding indication of sequential activity in the subset of neurons active during the maintenance period, we also identified different subsets that were active in encoding and retrieval periods and appeared to form sequences, too. Obviously, further work is needed to understand the population dynamics in this neuronal pool more precisely. We hope these comments clarified our intentions and conclusions.

Minor comments (Reviewer #3):

1. The authors could have been more thorough in characterizing the neurons they are targeting. In particular, it would have been useful to know what percentage of neurons in mPFC they are labeling. Additionally, since the authors are recording and manipulating cell bodies rather than terminals, it seems important to know whether the mPFC to dmStr neurons project to other areas besides dmStr. They briefly touch on this issue in Lines 449-452, but it would have been nice to show the full projection profile of these neurons or discuss any literature may have already done this. Alternatively, do the authors have any evidence that it's specifically the projection to the striatum which mediates the WM maintenance effects reported here?

Reply: We agree with the reviewer that further characterization of the subset of striatum-projecting mPFC neurons is desirable and that mPFC→dmStr projection neurons will have collaterals and connect to other regions, too. Some knowledge is available from the literature: In rats, approximately 18% of L5 neurons in the PL area project to dmStr (Gabbott et al 2005). Using double-labeling with retrograde tracers, the same study found about 2-3% of all PL L5 neurons projecting in a divergent manner to striatum and the basolateral amygdala, and about 1% double-projecting to dorsal striatum and spinal cord. A recent study also identified the pedunculo-pontine nucleus (PPN) as co-target of striatum-projecting PL neurons (Liu et al., 2022). We additionally checked the MouseLight project (Janelia Research Campus), where we found 4 reconstructed PL neurons that displayed projections to the dorsal striatum (both ipsi- and contralateral) and in addition to frontal cortical regions (anterior cingulate, orbitofrontal, frontal pole) and the insula. These divergent projection patterns match with our own 3D analysis of one cleared mouse brain with mPFC→dmStr projections labeled, where we found axons also in contralateral striatum, the insula, and in the internal capsula indicating projections towards the spinal cord. Overall, we consider these data as preliminary and rudimentary and therefore did not include them here. We believe that a thorough anatomical characterization will be needed to reveal the complete set of divergent projections and of target areas of mPFC→dmStr neurons.

We think that our results provide strong evidence for the importance of the mPFC→dmStr pathway in WM maintenance. We also like to point out that our electrophysiological recordings in striatum clearly showed an effect of optogenetic inhibition on this target region (Fig. 3c). Obviously, we cannot draw any conclusions about parallel effects on other target regions and their functional roles.

To at least mention this important point in the revised manuscript, we have now included a few sentences in the discussion (pg. 19, line 480-481).

2. The behavioral performance in Fig. 5 is highly variable across the different groups, making it difficult to fully accept the effects of opto-stimulation. For example, in panel 5d, left the mean performance is 60% both off and on laser. But in panel 5d, middle the mean performance is ~45% off laser and increases to ~65% on laser. Why was the off laser performance different?

Reply: We repeat here our response to point 1 of Reviewer #1: We addressed this question by adding three more experimental subjects to our dataset (3 additional mice with ChR2 activation, 6 more GFP control mice) and by performing the additional analysis described in our reply to General Concern 3. The baseline is now more homogenous across conditions of encoding, maintenance, and retrieval periods, with overlapping standard errors of the mean and indistinguishable variance across all conditions as assessed with Bartlett's test. We also treated Left Sample and Right Sample Runs separately.

We have updated Fig. 5c,d accordingly (see figure above; point 1 of reviewer #1) and now show the performance results for Left Sample runs in this main figure. For your information, we also showed the corresponding plot for Right Sample Runs, which does not show any significant differences as we now mention in the figure legend.

3. Page 3, Line 56: may be [better to say "interactions with MD nucleus in the thalamus..."]

Reply: We have modified the text according to the reviewer's suggestion.

We hope that we have convincingly explained our arguments and that based on our responses you may consider our manuscript for publication in Nature Communications.

Kind regards,

Yaroslav Sych, Maria Wilhelm, and Fritjof Helmchen

REVIEWERS' COMMENTS

Reviewer #1 (Remarks to the Author):

The authors have partially addressing some of my questions and concerns by performing additional (more detailed) behavioral analyses and by adding a few more animals (n=3) to the group in one experiment. These modifications are useful but not entirely satisfactory without additional experiments as outlined/requeusted in the previous comments.

Reviewer #2 (Remarks to the Author):

The authors have done a good job responding to prior reviews. I have no further major concerns. One minor one:

What criteria are used for clustering in Fig. S2c? UMAP can be calculated on any data, and k-means will cluster any data (given that you specify the number of clusters). It's not clear to me how this analysis adds to the study.

Reviewer #3 (Remarks to the Author):

The authors have carried out extensive revisions with new behavioral analyses and experiments. Their revised manuscript addresses all my previous concerns, and substantially supports their main claims. This study provides interesting and novel insights about the role of fronto-striatal circuits in working memory. I have no further concerns.

Responses to Reviewers' Comments, Nature Communications, NCOMMS-21-44417A

We again thank the reviewers for carefully reading our revised manuscript and for acknowledging the improvements that we have made in response to their concerns. Find below our responses to the remaining questions of the reviewers.

Responses to Reviewer #1:

The authors have partially addressing some of my questions and concerns by performing additional (more detailed) behavioral analyses and by adding a few more animals (n=3) to the group in one experiment. These modifications are useful but not entirely satisfactory without additional experiments as outlined/requested in the previous comments.

We appreciate that the reviewer found our additional results useful. We acknowledge that he/she would have liked to see further experiments, which for the stated reasons we unfortunately could not conduct.

Responses to Reviewer #2:

The authors have done a good job responding to prior reviews. I have no further major concerns. One minor one: What criteria are used for clustering in Fig. S2c? UMAP can be calculated on any data, and k-means will cluster any data (given that you specify the number of clusters). It's not clear to me how this analysis adds to the study.

We thank the reviewer for appreciating our revision work. Regarding the last minor concern about Fig. S2c, we like to emphasize that clustering here was meant to reveal distinct behavior types but rather to provide bins along the spectrum of motor behaviors (reflecting difference in behavioral features as analyzed in Fig. S2e). We then tested whether the corresponding photometry signals for the 3 considered behavioral bins differed, which was not the case (Fig. S2f,g). We take this result as confirmation that no obvious relationship between mPFC→dmStr pathway activity (as measured by photometry) and ongoing motor behavior could be detected, addressing a major concern by Reviewer #3.

We clarified our point of view in the main text (middle of page 6), which now reads:

"We also analyzed multiple behavioral variables but did not find clear clusters of distinct behaviors, rather a spectrum of behaviors. To evaluate whether variability of mPFC→dmStr activity during the maintenance period could be explained by this behavioral repertoire, we analyzed mPFC→dmStr activity at the extremes of the behavioral spectrum but did not find any obvious relationship between activity and the ongoing motor behavior (Supplementary Fig. 2c-g)."

Responses to Reviewer #3:

The authors have carried out extensive revisions with new behavioral analyses and experiments. Their revised manuscript addresses all my previous concerns, and substantially supports their main claims. This study provides interesting and novel insights about the role of fronto-striatal circuits in working memory. I have no further concerns.

We thank the reviewer for appreciating our revision work and for highlighting the novelty and potential impact of our research results.

Kind regards,

Yaroslav Sych, Maria Wilhelm, and Fritjof Helmchen